# Diffusion-Based Hierarchical Graph Neural Networks for Simulating Nonlinear Solid Mechanics

**Tobias Würth**[1]* **Niklas Freymuth**[2] **Gerhard Neumann**[2] **Luise Kärger**[1]

[1]Institute of Vehicle System Technology, Karlsruhe Institute of Technology, Karlsruhe
[2]Autonomous Learning Robots, Karlsruhe Institute of Technology, Karlsruhe

## Abstract

Graph-based learned simulators have emerged as a promising approach for simulating physical systems on unstructured meshes, offering speed and generalization across diverse geometries. However, they often struggle with capturing global phenomena, such as bending or long-range correlations usually occurring in solid mechanics, and suffer from error accumulation over long rollouts due to their reliance on local message passing and direct next-step prediction. We address these limitations by introducing the Rolling Diffusion-Batched Inference Network (ROBIN), a novel learned simulator that integrates two key innovations: (i) Rolling Diffusion-Batched Inference (ROBI), a parallelized inference scheme that amortizes the cost of diffusion-based refinement across physical time steps by overlapping denoising steps across a temporal window. (ii) A Hierarchical Graph Neural Network built on algebraic multigrid coarsening, enabling multiscale message passing across different mesh resolutions. This architecture, implemented via Algebraic-hierarchical Message Passing Networks, captures both fine-scale local dynamics and global structural effects critical for phenomena like beam bending or multi-body contact. We validate ROBIN on challenging 2D and 3D solid mechanics benchmarks involving geometric, material, and contact nonlinearities. ROBIN achieves state-of-the-art accuracy on all tasks, substantially outperforming existing next-step learned simulators while reducing inference time by up to an order of magnitude compared to standard diffusion simulators.

## 1 Introduction

Physical simulations enable many engineering and scientific fields to gain quick insights into complex systems or to evaluate design decisions. Conventional simulations model the physical system using Partial Differential Equations (PDEs). Usually, the PDE is discretized by numerical methods, such as the Finite Element Method (FEM) [1], the Finite Volume Method (FVM) [2], or the Finite Difference Method (FDM) [3]. This process reduces the need for cumbersome, resource-intensive real-world experiments. Recent research aims to speed up simulation with Machine Learning (ML)-based models [4, 5]. These learned simulators promise to allow researchers and practitioners to evaluate large amounts of virtual, simulated experiments. These simulations in turn unlock applications in engineering design and manufacturing optimization [6–8].

This work aims to improve learned simulators, focusing on simulations of nonlinear solid mechanics as a representative class of examples. We combine recent image-based Denoising Diffusion Probabilistic Models (DDPMs) [9–11] with Hierarchical Graph Neural Networks (HGNNs) [12–14] (cf. Figure 1). While diffusion has shown promising results on images [15–17], audio [18, 19] and even policy learning for robotics [20, 21], it suffers from cost-intensive inference due to its iterative denoising procedure. We alleviate this high inference cost on time-dependent domains with Rolling Diffusion-

---

*correspondence to `tobias.wuerth@kit.edu`

39th Conference on Neural Information Processing Systems (NeurIPS 2025).

Batched Inference (ROBI), a novel scheduling scheme that batches denoising steps of consecutive time steps. ROBI already starts denoising future prediction steps by using partially refined previous steps. This process reduces the number of model evaluations to the number of time steps and preserves the time-shift equivariance of Markovian systems. ROBI only affects the inference process, allowing us to utilize conventional, parallelized DDPM training.

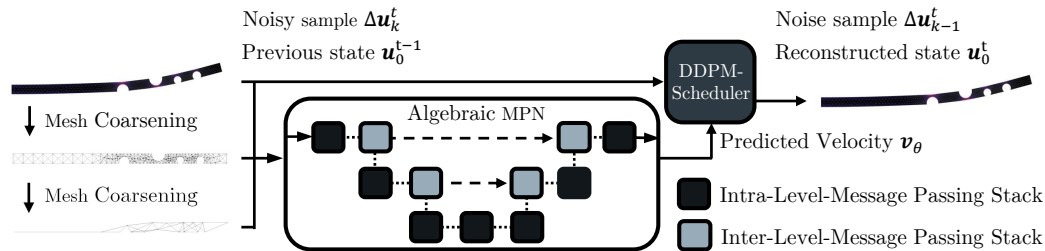

Figure 1: Overview of a Rolling Diffusion-Batched Inference Network (ROBIN) prediction. ROBIN coarsens the fine mesh multiple times with Algebraic multigrid (AMG) to create a graph hierarchy. ROBIN predicts the denoising velocity $\mathbf{v}_\theta(\Delta\mathbf{u}_k^t, k, \mathbf{u}_0^{t-1})$ at time step $t$ using Algebraic-hierarchical Message Passing Networks (AMPNs), given a noisy residual sample $\Delta\mathbf{u}_k^t$, the diffusion step $k$, and a previous state $\mathbf{u}_0^{t-1}$. The prediction is used to draw a new noisy residual sample $\Delta\mathbf{u}_{k-1}^t$ and to update the state $\mathbf{u}_0^t$.

We combine ROBI with DDPMs and Algebraic-hierarchical Message Passing Networks (AMPNs) to form the Rolling Diffusion-Batched Inference Network (ROBIN), which significantly accelerates simulation while improving predictive accuracy. ROBIN constitutes the first diffusion-based refiner for simulating physical dynamics on unstructured meshes, surpassing the current accuracy ceiling of HGNNs. We train ROBINs on three challenging 2D and 3D solid mechanics datasets involving geometric, material, and contact nonlinearities. Across all datasets, ROBIN significantly improves over state-of-the-art mesh-based simulators [5, 22] in terms of predictive accuracy. Leveraging DDPMs, ROBIN accurately captures low-frequency global solution modes while resolving high-frequency components. We further find that our proposed inference method, ROBI, speeds up diffusion-based inference for learned simulations by up to an order of magnitude while maintaining accuracy. In addition, the AMPN architecture of ROBIN enables efficient transfer to much larger meshes, while maintaining near-FEM accuracy.[2]

To summarize, we i) propose ROBI, a novel inference scheduling scheme for diffusion-based simulators that amortizes denoising across time steps, reducing inference to a single model evaluation per step while preserving time-shift equivariance; ii) introduce ROBIN, a diffusion-based HGNN for nonlinear solid mechanics that combines multiscale message passing with ROBI to provide fast, accurate diffusion-based simulations; iii) demonstrate state-of-the-art performance on challenging 2D and 3D benchmarks, outperforming existing simulators in both accuracy and runtime.

## 2 Related Work

**Simulating Complex Physics.** Simulating complex physical systems often requires numerical solvers, such as the FEM [23, 24, 1], the FVM [2], or the FDM [3]. While accurate, numerical solvers scale poorly with problem complexity, often requiring multiple hours or even days for a single rollout on a modern workstation. Recent work shows that ML-based models are able to learn such numerical simulations from data [25, 26, 4, 5, 27, 9, 22, 10]. ML-based models provide speed-ups of one to two orders of magnitude while being fully differentiable, which accelerates downstream applications such as design [6] or manufacturing process optimization [7, 8]. Many learned simulators operate autoregressively. They mimic numerical solvers by using their own predictions to estimate the residual between successive time steps [4, 5, 28]. Similarly, Neural ODEs [29, 30] predict time derivatives and advance solutions via numerical integration. In contrast to these supervised approaches, Physics-Informed Neural Networks [31] directly operate on a PDE loss function to train Multilayer Perceptrons (MLPs) [32, 33, 8], Convolutional Neural Networks (CNNs) [34, 35] or Graph Neural Networks (GNNs) [36, 37]. Finally, Neural Operators aim to learn mesh-independent solution operators [38–40, 27, 41]. ROBIN is an autoregressive learned simulator that replaces direct

---

[2]Project page, code and datasets are available at https://tbswrth.github.io/ROBIN.

next-step prediction with multiple denoising diffusion steps, leveraging generative inference to improve prediction accuracy.

**Learning Mesh-based Simulations with Graph Neural Network (GNN).** Pfaff et al. [5] introduced MESHGRAPHNETS, a Message Passing Network (MPN)-based simulator that encodes simulation states as graphs using mesh connectivity and physical proximity. While accurate for small problems, MESHGRAPHNETS (MGNs) does not scale well, as its receptive field is limited by the number of message-passing steps in the MPN. To address this issue, recent work expands the receptive field via global attention [42–44] or hierarchical mesh representations using Graph Convolutional Networks [45] and MPNs [46, 13]. Extensions further improve efficiency and accuracy through rotation equivariance [14], hierarchical edge design [47], bi-stride pooling [48], and attention mechanisms acting on edges and across hierarchies [22]. [49] leverages Adaptive Mesh Refinement (AMR) to create mesh hierarchies for multi-scale GNNs. Complementary, Physics-Informed MESHGRAPH-NETS [37] integrate FEM-based training to improve accuracy and robustness. Existing methods are generally trained using a next-step Mean Squared Error (MSE) loss, which favors learning lower solution frequencies at the cost of accuracy in higher frequency bands that have less impact on the loss [9]. However, autoregressive models trained with a MSE loss often overlook low-amplitude frequencies [9]. Our approach is orthogonal, coupling hierarchical GNN with denoising diffusion models. This approach takes advantage of the large receptive field of multi-scale GNNs while pushing accuracy toward diffusion limits.

**Diffusion-based Simulations.** Diffusion models have been applied to physics-informed image super-resolution [35], flow field reconstruction [50], and steady-state flow generation on grids using CNNs [51], and more recently to meshes with hierarchical GNNs [52]. These models, however, operate on isolated frames and do not capture time-dependent dynamics. In contrast, our model predicts deterministic physical evolution rather than equilibrium samples via autoregressive rollouts.

While next-step simulators trained with MSE loss capture high-amplitude, low-frequency components, they often miss low-amplitude components [9]. PDERefs (PDE-Refiners) address this via iterative refinement, improving long-horizon accuracy of grid-based CNN simulators. We extend this idea to unstructured meshes by combining algebraic mesh coarsening [53, 54] with hierarchical GNNs [52, 22] with shared layers. We further introduce a time-parallel denoising scheme at inference, removing the speed bottleneck while maintaining accuracy. Unlike diffusion-based CNN simulators that require $K$ model evaluations per physical step [11, 55, 9], our method requires only a single hierarchical GNN call per step post warm-up, reducing inference costs over $T$ time steps from $\mathcal{O}(KT)$ to $\mathcal{O}(T)$. Compared to video-based approaches [10, 56], which need to jointly process $N$ steps and learn a time-dependent denoiser with high memory cost, our model, ROBIN, leverages time-translation invariance to train a time-independent denoiser with only one time step in memory. As such, ROBIN can be applied autoregressively and can freely interpolate between fully parallel denoising and memory-efficient one-step denoising at inference time. It also predicts state residuals instead of states, significantly improving long-horizon rollout fidelity.

## 3   Rolling Diffusion-Batched Inference Network (ROBIN)

**Graph Network Simulators (GNSs) for Mesh-based Simulations.** We consider solving PDEs for physical quantities $\mathbf{u}(\mathbf{x}, t)$ that change over time $t \in [0, T]$ and inside a time-dependent domain $\mathbf{x}(t) \in \Omega(t)$. We focus on simulations on meshes, where $\mathcal{G} = (\mathcal{V}, \mathcal{E}^{\mathrm{M}})$ denotes the mesh graph and the graph nodes $\mathcal{V}$ and the graph edges $\mathcal{E}^{\mathrm{M}}$ correspond to mesh nodes and mesh edges. We seek solutions $\mathbf{u}_i(t) = \mathbf{u}(\mathbf{x}_i, t)$ at discrete node locations $\mathbf{x}_i(t) \in \Omega(t)$. To obtain discretized PDEs, usually numerical methods, such as the FEM, are applied that define the discretization of spatial operators, such as gradients $\partial \mathbf{u}(\mathbf{x}, t)/\partial \mathbf{x}$. Given the discretized operator $\mathcal{F}$, the PDE simplifies to a time-dependent Ordinary Differential Equation and requires solving $\partial \mathbf{u}_i/\partial t = \mathcal{F}(t, \mathbf{x}_i, \mathbf{u}_i)$. We can solve such systems using numerical time discretization schemes. In this work we use a simple *Euler* forward discretization $\mathbf{u}_i^{t+1} = \mathbf{u}_i^t + \Delta t\, \mathcal{F}(t, \mathbf{x}_i^t, \mathbf{u}_i^t)$, and set $\Delta t = 1$. We extend PDE-Refiner [9] to *Lagrangian* systems, where the domain $\Omega^t$ and node locations $\mathbf{x}_i^t \in \Omega^t$ evolve over time. Here, we predict the solution $\mathbf{u}_i^t$ at time step $t$ by learning to reverse a probabilistic diffusion process [57] conditioned on the previous state of time step $t-1$. The proposed methods also apply without any restriction to *Eulerian* systems, where the domains are time-independent.

## 3.1 Denoising Diffusion Probabilistic Models (DDPMs) for time-dependent simulations

Given a time-dependent solution $\mathbf{u}^t$ from a data distribution $q(\mathbf{u})$, the forward diffusion process is modeled by a Markov chain, where Gaussian noise is added gradually to the sample $\mathbf{u}_k^t$ at each diffusion step $k$

$$q(\mathbf{u}_{1:K}^t | \mathbf{u}_0^t) = \prod_{k=1}^{K} q(\mathbf{u}_k^t | \mathbf{u}_{k-1}^t) \,, \quad q(\mathbf{u}_k^t | \mathbf{u}_{k-1}^t) := \mathcal{N}(\mathbf{u}_k^t; \sqrt{1 - \beta_k} \mathbf{u}_{k-1}^t, \beta_k \mathbf{I}) \,.$$

$\mathcal{N}$ denotes the normal distribution and $\beta_k$ the noise specified by a variance scheduler. We learn to reverse the diffusion process, i.e.,

$$p_\theta(\mathbf{u}_{k-1}^t | \mathbf{u}_k^t) := \mathcal{N}(\mathbf{u}_{k-1}^t; \mu_\theta(\mathbf{u}_k^t, k), \Sigma(\mathbf{u}_k^t, k)) \,,$$

consisting of $K$ iterative diffusion steps and starting from $k = K$. The mean $\mu_\theta$ depends on the prediction of a learned model with parameters $\theta$. The covariance $\Sigma$ is assumed to be isotropic and given as $\Sigma = \sigma_k^2 \mathbf{I} = (\frac{1 - \bar{\alpha}_{k-1}}{1 - \bar{\alpha}_k} \beta_k) \mathbf{I}$ [57] with $\alpha_i = 1 - \beta_i$ and $\bar{\alpha}_k = \prod_{i=1}^{k} \alpha_i$. We train the model to predict the denoising velocity, i.e., the *v-prediction target*

$$\mathbf{v}_k^t = \sqrt{\bar{\alpha}_k} \epsilon^t - \sqrt{1 - \bar{\alpha}_k} \mathbf{u}_0^t \,, \tag{1}$$

given gaussian noise $\epsilon^t$ [58]. Since this target smoothly interpolates between $-\mathbf{u}_0^t$ as $\bar{\alpha}_k \approx 0$ and $\epsilon^t$ ($\bar{\alpha}_k \approx 1$), it emphasizes predicting the clean sample in early (high-noise) steps and the noise in later (high-signal) steps, simplifying learning. The model predicts the denoising velocity $\mathbf{v}_\theta(\mathbf{u}_k^t, k) = \mathbf{v}_\theta(\mathbf{u}_k^t, k, \mathbf{u}_0^{t-1})$ autoregressively, conditioned the model on the last time step solution $\mathbf{u}_0^{t-1}$. The training objective is then defined as $\mathbb{E}_{\mathbf{u}_0^t, \epsilon^t, k} \left[ \left\| \mathbf{v}_\theta(\mathbf{u}_k^t, k, \mathbf{u}_0^{t-1}) - \mathbf{v}_k^t \right\|^2 \right]$ [9]. To facilitate faster denoising, we follow the DDPM formulation of [9] and use an exponential $\beta_k$ scheduler.

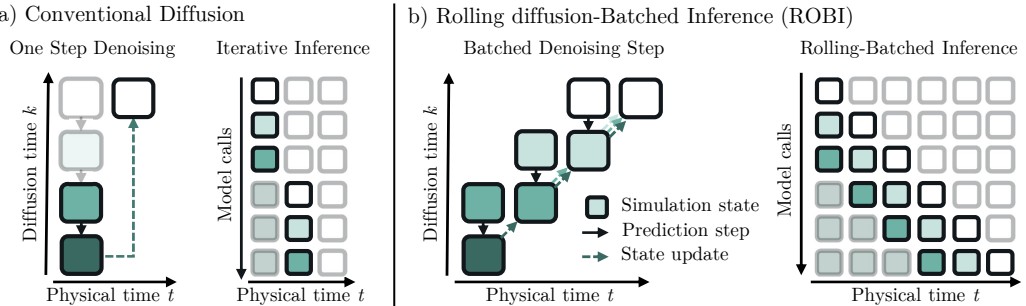

Figure 2: Overview of **a)** conventional autoregressive diffusion inference and **b)** ROBI. **a)** Conventional inference denoises the entire state of a physical time step at once before it shifts to the next time step (see *One Step Denoising*). The *Iterative Inference* requires $K$ model calls per time step, where $K$ denotes the number of diffusion steps. **b)** ROBI parallelizes denoising steps across physical time, processing up to $K$ time steps batched, and reconstruct the physical states with the clean sample prediction subsequently (see *Batched Denoising Step*). This process allows *Rolling-Batched Inference* after the initial warm-up, reducing the number of model calls to one per time step.

The first denoising step is defined such that $\bar{\alpha}_K \approx 0$, which simplifies the v-prediction target to $\mathbf{v}_k^t \approx -\mathbf{u}_0^t$ (cf. Equation (1)). The noisy sample $\mathbf{u}_k^t \approx \epsilon^t$ corresponds to Gaussian Noise and is uninformative. Hence, for $k = K$ the model target converges to $\left\| \mathbf{v}_\theta(\mathbf{u}_K^t, K, \mathbf{u}_0^{t-1})) - \mathbf{v}_k^t \right\|^2 \approx \left\| \mathbf{v}_\theta(\epsilon^t, K, \mathbf{u}_0^{t-1})) + \mathbf{u}_0^t \right\|^2$. This MSE objective of the first denoising step $k = K$ mirrors the training objective of one-step models [9], i.e., auto-regressive models that predict the solution of the next time directly with a single prediction step. Hence, we expect similar accuracy as one-step models during the first diffusion steps. However, the DDPM-based model refines the initial prediction iteratively at each time step, improving the accuracy of the solution further. Note that, due to the noise scheduler,

each denoising step focuses on different amplitude and frequency levels of the solution [9], with later denoising steps increasingly paying attention to higher frequencies.

**Rolling Diffusion-Batched Inference (ROBI).** Conventional diffusion inference requires $K$ model calls, each corresponding to a denoising step, per simulation time step [9]. Figure 2 a) shows an example. Thus, inference is roughly $K$ times slower than one-step models [5]. We propose Rolling Diffusion-Batched Inference (ROBI) to accelerate inference in DDPM-based autoregressive simulators. Given a velocity prediction $\mathbf{v}_\theta(\mathbf{u}_k^t, k, \mathbf{u}_0^{t-1})$ at the denoising step $k$, we reconstruct a partially refined prediction $\tilde{\mathbf{u}}_{0|k}^t = \sqrt{\bar{\alpha}_k}\mathbf{u}_k^t - \sqrt{1 - \bar{\alpha}_k}\mathbf{v}_\theta(\mathbf{u}_k^t, k, \mathbf{u}_0^{t-1})$, following [58].

As discussed in Section 3.1, early denoising steps behave similarly to one-step model predictions, while later steps progressively refine higher spatial frequencies (from coarse structures to fine details). After $m < K$ denoising steps at time $t$, the intermediate estimate $\tilde{\mathbf{u}}_{0|m}^t$ already captures low-to-mid frequency content that is sufficient to condition the next physical step $t+1$ to predict a solution within this already refined frequency band.

ROBI exploits this property by starting the denoising of step $t+1$ as soon as step $t$ has progressed by $m$ steps. We initialize $\mathbf{u}_K^{t+1} \sim \mathcal{N}(\mathbf{0}, \mathbf{I})$ and denoise the batch of both time steps $t_w \in \{t, t+1\}$. More generally, after $jm$ denoising steps, we initialize the time step $t+j$ and denoise a rolling time window with $t_w \in \{t, ..., t+j\}$ in parallel. After a short warm-up, fully denoised samples (those with $k=0$) drop out and the window size becomes constant with $w=K/m$ and $t_w \in \{t, ..., t+w-1\}$. Notably, the denoising index $k$ and thus the noise level increase along the window toward later physical times, aligning with the natural growth of forecast uncertainty. This reduces the number of model calls from $KT$ (conventional inference) to $K-m+mT$ steps. For $m=1$, the number of model calls reduces to $K+T$, which is effectively the same complexity $\mathcal{O}(T)$ as for one-step models, when $K \ll T$. We refer to $m$ as the *denoising stride*. Figure 2 b) visualizes the special case $m=1$ and $K=3$ diffusion steps. Each model call advances the simulation by one physical step and applies one denoising step to each of the $K$ partially denoised predictions in the window.

A single *Batched Denoising Step* (cf. Figure 2 b) left for $m=1$ and $K=3$) of ROBI consists of two consecutive steps. In the *Prediction step*, the model outputs the velocity $\mathbf{v}_\theta(\mathbf{u}_{k_w}^{t_w}, k_w, \mathbf{u}_0^{t_w-1})$ for each element in the prediction window $t_w \in \{t, ..., t+w-1\}$, with the diffusion indices $k_w = \in \{m, ..., K\}$. Subsequently, we perform a *State Update* step. Using the scheduler, we reconstruct $\tilde{\mathbf{u}}_{0|k_w}^{t_w}$ to update the conditioning states $\mathbf{u}_0^{t_w}$, and sample the reverse transition $\mathbf{u}_{k_w-1}^{t_w} \sim p_\theta(\mathbf{u}_{k_w-1}^{t_w}|\mathbf{u}_{k_w}^{t_w})$. Both are used in the next *Prediction Step* as inputs. After $m$ such *Batched Denoising Steps*, we advance in time and drop the fully denoised states ($k=0$) and initialize a new Gaussian sample ($k=K$) for the next time index $t+w$, so the subsequent call evaluates $\mathbf{v}_\theta(\mathbf{u}_{k_w}^{t_w+1}, k_w, \mathbf{u}_0^{t_w})$.

For $m=K$, ROBI reduces to conventional autoregressive diffusion inference (cf. Figure 2 a)). Thus, the denoising stride $m$ can be considered a hyperparameter that trades off prediction accuracy and memory usage with inference speed. Most notably, in *State Update*, ROBI reconstructs the physical states, while the rolling time window is treated purely as a batch dimension in the prediction model. Consequently, the model always predicts $\mathbf{v}_\theta(\mathbf{u}_k^t, k) = \mathbf{v}_\theta(\mathbf{u}_k^t, k, \mathbf{u}_0^{t-1})$, i.e., conditioned only on the previous reconstructed state during both training and inference, which proves to be more stable and accurate for autoregressive ML-based simulations [5, 9]. Furthermore, this preserves the time-shift equivariance of Markovian systems, fast training convergence and small GPU memory utilization.

In practice, we find predicting residuals $\Delta\mathbf{u}_k^t$ improves accuracy. Let the model denoise a batch of residual states $\Delta\tilde{\mathbf{u}}_0^{t_w} \approx \Delta\tilde{\mathbf{u}}_{0|k_w}^{t_w}$ of the time window $t_w \in \{t, ..., t+w-1\}$ and denote the last fully denoised state as $\mathbf{u}_0^{t-1}$. We then recover clean states inside the window via a cumulative sum $\tilde{\mathbf{u}}_0^{t+j} = \mathbf{u}_0^{t-1} + \sum_{i=0}^{j} \Delta\tilde{\mathbf{u}}_0^{t+i}$, $j \in \{0, ..., w-1\}$. To further accelerate inference, we optionally stop denoising early at a *truncation step* $k_{\text{tr}}$ and use the partially denoised state $\tilde{\mathbf{u}}_{0|k_{\text{tr}}}^t$ as the final prediction of the time step. This remains effective because early denoising steps already approximate one-step prediction.

## 3.2 Denoising Diffusion Probabilistic Models (DDPMs) for mesh-based simulations

To fully utilize Denoising Diffusion Probabilistic Models (DDPMs)'s potential for generating rich, multi-frequency solutions, prediction models must handle multi-scale information. Hierarchical Graph Neural Networks (HGNNs) are particularly well-suited for this, as their architecture inherently

learns representations at varying levels of granularity, mirroring the diverse frequency content present in DDPM outputs. Leveraging this idea, we train HGNN to predict the discrete denoising velocity $\mathbf{v}_{i,\theta}(\mathbf{u}_{i,k}^t, k, \mathbf{u}_{i,0}^{t-1})$ for mesh-based simulations on the mesh graph $\mathcal{G} = (\mathcal{V}, \mathcal{E}^M)$. It takes the current noisy sample $\mathbf{u}_{i,k}^t$ and is conditioned on the previous clean sample $\mathbf{u}_{i,0}^{t-1}$.

**Root-node AMG-based Mesh Coarsening.** We construct a hierarchical mesh graph $\mathcal{G}^H = \mathcal{G}^{0:L} = (\mathcal{V}^{0:L}, \mathcal{E}^{0:L,M})$ consisting of $L+1$ mesh graphs with nodes $\mathcal{V}^{0:L}$ and mesh edges $\mathcal{E}^{0:L,M}$ by coarsening the fine graph $\mathcal{G}^0 := \mathcal{G}$ $L$ times. Coarsening is performed with root-node smoothed aggregation [53] implemented in [59]. The solver takes the fine-mesh adjacency (with self-loops) $A^0$ as its system matrix and creates a hierarchy of adjacency $A^{0:L}$, upsampling (prolongation) $U^{0:L-1}$ and downsampling (restriction) $D^{0:L-1}$ matrices. In practice, our implementation only requires the fine adjacency $A^0$, the root-node solver [59], and constructing the graphs of each level from the non-zeros of the returned matrices $A^{1:L}$. The resulting nodes $j \in \mathcal{V}^{0:L}$ of each level are always a subset of the fine mesh nodes. Unlike bi-stride pooling with Delaunay remeshing [22], this AMG-based approach better preserves mesh geometry by leveraging the strength of connections in the adjacency matrix. Figure 10 in Section E visualizes this difference.

We additionally define $L-1$ downsampling edges $\mathcal{E}^{l,D}$ and upsampling edges $\mathcal{E}^{l,U}$, which connect nodes between successive levels $\mathcal{G}^l$ and $\mathcal{G}^{l+1}$. We define these as the connections (non-zero values) of the given up- $U^{0:L-1}$ and downsampling $D^{0:L-1}$ matrices, resulting in an extended hierarchy graph $\mathcal{G}^H = (\mathcal{V}^{0:L}, \mathcal{E}^{0:L,M} \cup \mathcal{E}^{0:L-1,D} \cup \mathcal{E}^{0:L-1,U})$. As before, we found that the obtained pooling mappings respect the mesh geometry well, as shown in Figure 10 d) in Section E.

For contact experiments, we add contact edges $\mathcal{E}^{0,C}$ [22] to the graph hierarchy $\mathcal{G}^H$. In a simulation involving two colliding components, we define a bidirectional edge between the nodes of the first part and the nodes of the second part if their distance is less than the specified contact radius $R$. The resulting hierarchy is given by $\mathcal{G}^H = (\mathcal{V}^{0:L}, \mathcal{E}^{0:L,M} \cup \mathcal{E}^{0:L,C} \cup \mathcal{E}^{0:L-1,D} \cup \mathcal{E}^{0:L-1,U})$.

### 3.3 Algebraic-hierarchical Message Passing Networks (AMPNs).

**Encoder.** Let $\mathcal{G}^H$ be a hierarchical graph as defined above and $\mathbf{u}_{i,k}^t$ the noisy sample at denoising step $k$ of simulation step $t$. We define node embeddings $\mathbf{k}_i \in \mathcal{V}^{0:L}$, mesh edge embedding $\mathbf{e}_{ij}^M \in \mathcal{E}^{0:L,M}$, contact edge embeddings $\mathbf{e}_{ij}^C \in \mathcal{E}^C$, downsampling edge embeddings $\mathbf{e}_{ij}^D \in \mathcal{E}^{0:L-1,D}$ and upsampling edge embeddings $\mathbf{e}_{ij}^U \in \mathcal{E}^{0:L-1,U}$. We add relative node distances $\mathbf{x}_{ij}^t = \mathbf{x}_i^t - \mathbf{x}_j^t$ and their norm $|\mathbf{x}_{ij}^t|$ to all edge embeddings and the initial distance $\mathbf{x}_{ij}^0 = \mathbf{x}_i^0 - \mathbf{x}_j^0$ and their norm $|\mathbf{x}_{ij}^0|$ to mesh edges, down- and upsampling embeddings. Node embeddings include a one-hot encoding $n_i$ of the node type. Node embeddings at level $l = 0$ additionally include $\mathbf{u}_{i,k}^t$ and task-specific features. All embeddings are projected to the latent dimension $d$ via linear layers. We add a Fourier encoding [57] for the denoising step $k$ and the normalized level $l^* = l/L$ to inform the AMPN of relative graph depth.

**Processor and Decoder.** Similar to AMG solvers [54, 53] and UNets [60], Algebraic-hierarchical Message Passing Networks (AMPNs) use a *V-cycle* to propagate information between levels. They consist of five core message passing modules: Pre-Processing, Downsampling, Solving, Upsampling and Post-Processing, as shown in Figure 1. Pre-Processing, Solving and Post-Processing modules use an Intra-Level-Message Passing Stack (Intra-MP-Stack) consisting of $N$ message passing steps to update the heterogeneous subgraph $\mathcal{G}^l = (\mathcal{V}^l, \mathcal{E}^{l,C} \cup \mathcal{E}^{l,M})$ of level $l$. Given node embeddings $\mathbf{k}_i \in \mathcal{V}^l$, contact edge embeddings $\mathbf{e}_{ij}^C \in \mathcal{E}^{l,C}$ and mesh edge embeddings $\mathbf{e}_{ij}^M \in \mathcal{E}^{l,M}$, the message passing update of the level graph at step $n$ is defined by

$$
\begin{aligned}
\mathbf{e}_{ij}^{C,n+1} &= W_{\theta,\mathcal{E}^C}^n \, \mathbf{e}_{ij}^{C,n} + f_{\theta,\mathcal{E}^C}^n(\mathbf{k}_i^n, \mathbf{k}_j^n, \mathbf{e}_{ij}^{C,n}) \,, \\
\mathbf{e}_{ij}^{M,n+1} &= W_{\theta,\mathcal{E}^M}^n \, \mathbf{e}_{ij}^{C,n} + f_{\theta,\mathcal{E}^M}^n(\mathbf{k}_i^n, \mathbf{k}_j^n, \mathbf{e}_{ij}^{M,n}) \,, \\
\mathbf{k}_i^{n+1} &= W_{\theta,\mathcal{V}}^n \, \mathbf{k}_i^n + f_{\theta,\mathcal{V}}^n(\mathbf{k}_i^n, \bigoplus_j \mathbf{e}_{ij}^{C,n+1}, \bigoplus_j \mathbf{e}_{ij}^{M,n+1}) \quad .
\end{aligned}
\tag{2}
$$

The operator $\bigoplus$ denotes a permutation-invariant aggregation, $f_{\theta,\cdot}^n$ MLPs and $W_{\theta,\cdot}^n$ weight matrices [46, 5, 52]. Downsampling modules update the subgraph $\mathcal{G}^{l,D} = (\mathcal{V}^{l+1} \cup \mathcal{V}^l, \mathcal{E}^{l,D})$ with an Inter-Level-Message Passing Stack (Inter-MP-Stack) of $N$ message passing steps. The receiver embeddings are

$\mathbf{k}_i^{\text{rec}} \in \mathcal{V}^{l+1}$, the sender embeddings $\mathbf{k}_j^{\text{send}} \in \mathcal{V}^l$, and the edge embeddings $\mathbf{e}_{ij} \in \mathcal{E}^{l,\text{D}}$. Similarly, the upsampling layers update the embeddings of the subgraph $\mathcal{G}_l^{\text{U}} = (\mathcal{V}^{l+1} \cup \mathcal{V}^l, \mathcal{E}^{l,\text{U}})$ of level $l$. Here, the receiver embeddings are $\mathbf{k}_i^{\text{rec}} \in \mathcal{V}^l$, the sender embeddings $\mathbf{k}_j^{\text{send}} \in \mathcal{V}^{l+1}$, and the edge embeddings $\mathbf{e}_{ij} \in \mathcal{E}^{l,\text{U}}$. A message passing step of an Intra-MP-Stack is defined as

$$
\begin{aligned}
\mathbf{e}_{ij}^{n+1} &= W_{\theta,\mathcal{E}}^n \, \mathbf{e}_{ij}^n + f_{\theta,\mathcal{E}}^n(\mathbf{k}_i^{\text{rec},n}, \mathbf{k}_j^{\text{send},n}, \mathbf{e}_{ij}^n) , \\
\mathbf{k}_i^{\text{rec},n+1} &= W_{\theta,\mathcal{V}}^n \, \mathbf{k}_i^{\text{rec},n} + f_{\theta,\mathcal{V}}^n(\mathbf{k}_i^{\text{rec},n}, \bigoplus_j \mathbf{e}_{ij}^{n+1}) \quad .
\end{aligned}
\tag{3}
$$

Our *V-cycle* starts at level $l = 0$ with pre-processing and downsampling at each level, repeated until the coarsest level $l = L$ is reached. We then apply multiple message passing steps at level $L$, which has the largest receptive field. Next, we upsample and post-process each level back up to $l = 0$. All Pre-Processing, Downsampling, Upsampling, and Post-Processing modules share weights across levels. A final linear layer decodes the fine-level node embeddings $\mathbf{k}_i \in \mathcal{V}^0$ to produce the velocity prediction $\mathbf{v}_{i,\theta}(\mathbf{u}_{i,k}^t, k, \mathbf{u}_{i,0}^{t-1})$.

## 4 Experiments

**Datasets.** We evaluate our model on the three different datasets, namely BENDINGBEAM, IMPACT-PLATE [22] and DEFORMINGPLATE [5]. We introduce the BENDINGBEAM dataset (Figure 3a)), featuring quasi-static, geometrically non-linear deformations of beams with high aspect ratios. The setup challenges models to capture global deformations via broad receptive fields and resolve high spatial frequencies due to locally thin, low-stiffness regions. In IMPACTPLATE (Figure 3 b)), the models must learn flexible dynamics with varying material parameters and accurately resolve very localized deformation at the contact point. DEFORMINGPLATE (Figure 3 c)) considers quasi-static contact simulations induced by scripted actuators that deform 3D plates consisting of nonlinear, hyperelastic material.

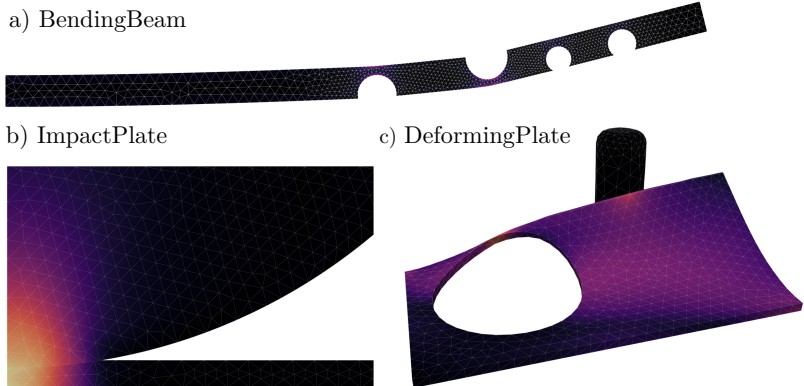

a) BendingBeam

b) ImpactPlate      c) DeformingPlate

Figure 3: Example predictions of ROBIN on the considered datasets. ROBIN predicts the part deformations as well as the von Mises stress (color, yellow is high) on all experiments. **a)** BENDINGBEAM considers global part deformations of beams induced by local forces. **b)** In IMPACTPLATE, the models have to predict locally large deformations, caused by a collision with an accelerated ball. **c)** The hyperelastic plates in DEFORMINGPLATE are deformed by a scripted actuator.

**Experimental setup.** For all tasks, we target the displacement residual of the node positions with respect to the next time step $\Delta\mathbf{u}_{i,0}^t = \mathbf{x}_i^{t+1} - \mathbf{x}_i^t$ during denoising. We additionally denoise the von Mises stress $\sigma_{\text{vMises},i,0}^t$ directly without residuals to gain further insight into the dynamics of the three experiments. ROBIN uses $K = 20$ denoising steps and a denoising stride of $m = 1$ by default. The task-specific features are specified in Section C, listed in Table 2. We measure the prediction error using an *RMSE*, as specified in Section C. Section C also provides information about additional settings of ROBIN, including training details and hyperparameters.

**Baselines.** We compare our model with three prominent baselines for nonlinear deformation simulations, namely MGNs, Hierarchical Contact Mesh Transformers (HCMTs), and Bi-Stride Multi-Scale GNNs (BSMSs). Detailed setups are provided in Section D.

**Variants.** We demonstrate that a *single trained* ROBIN can easily switch between different rollout modes by varying the denoising stride $m$ and the truncation step $k_{tr}$. We therefore evaluate multiple rollout settings $(m/k_{tr})$ on the same trained model ROBIN model: the default (1/20), conventional autoregressive diffusion inference (20/20) (i.e., $m=20$ denoising steps per physical step), and an intermediate variant (5/20) with $m = 5$. To study early stopping, we further reduce the truncation step $k_{tr}$ with (1/10), (1/5), (1/3), (1/2) and (1/1).

**Ablations.** We ablate key components of ROBIN to assess their impact. *10 diffusion steps* and *5 diffusion steps* train with reduced diffusion length $K$. *W/o diffusion* trains the AMPN as a one-step autoregressive model with MSE loss. *W/o hierarchy* disables hierarchical message passing, operating only on the fine mesh $G^0$. *State prediction* replaces residual-based prediction with direct denoising of $\mathbf{u}_{i,k}^t$ instead of $\Delta\mathbf{u}_{i,k}^t$. *W/o shared layer* uses 15 unshared message passing layers. *HCMT model* replaces AMPNs by HCMTs as the hierarchical model for ROBIN. Section D provides additional implementation and training details.

**Generalization to large meshes.** To assess upscaling, we introduce BENDINGBEAMLARGE with meshes on average over ten times larger than in BENDINGBEAM. We compare fine-tuning a pre-trained ROBIN to training from scratch, showing rapid transfer to substantially larger meshes without architectural changes, enabled by shared blocks in the AMPNs that accommodate varying hierarchy depths.

## 5  Results

**Baselines.** Figure 4 a) compares the rollout RMSE of ROBIN against HCMT, MGN, and BSMS. ROBIN yields substantial error reductions for the prominent IMPACTPLATE and DEFORMING-PLATE datasets, and yields an even larger improvement on BENDINGBEAM. Figure 5 demonstrates that ROBIN is able to propagate local boundary conditions across the part for accurate predictions of the global part deformation. In addition, ROBIN resolves the non-linear bending curve of the FEM. This requires fusing geometric features across scales, e.g., global part dimensions together with local thin walls, for an accurate prediction of the global part stiffness. All baselines struggle to reproduce the FEM results, particularly the global deformation modes.

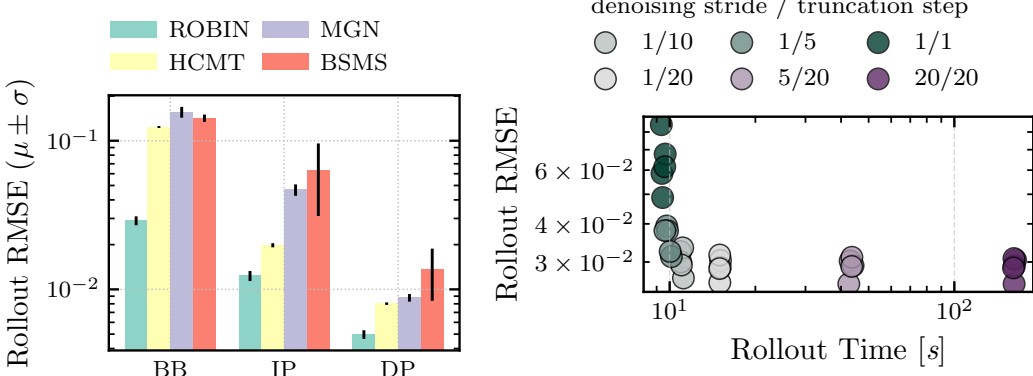

Figure 4: **Left:** Rollout error measured by the RMSE of the predicted nodes positions. ROBIN surpasses the accuracy of the baselines HCMT, MGN, and BSMS on all three datasets BENDING-BEAM, IMPACTPLATE and DEFORMINGPLATE. **Right:** Comparison of inference time and error of ROBIN and its variants on BENDINGBEAM. The default variant of ROBIN (1/20) achieves the same accuracy as conventional diffusion inference (20/20), while the inference speed is close to the one step variant (1/1). Reducing the truncation step $k_{tr}$ trades accuracy for speed.

**Inference speed.** Figure 4 b) plots rollout RMSE versus inference time on BENDINGBEAM. The default variant (1/20) of ROBIN is about an order of magnitude faster than conventional autoregressive diffusion inference (20/20), without compromising accuracy. Decreasing the truncation step $k_{tr}$ (i.e., fewer diffusion steps per physical step at inference) trades rollout accuracy for speed. Nevertheless, the fastest variant (1/1), which can be seen as a one-step model variant of ROBIN, still significantly outperforms the accuracy of the baselines (cf. Figure 4 a)). Most notably, ROBIN (1/20) requires only $\approx 58\%$ more time than the one-step variant (1/1), despite performing $K=20$ denoising steps per time step, highlighting the efficiency of the parallel denoising scheme ROBI. Similar trends hold for IMPACTPLATE and DEFORMINGPLATE (cf. Figure 11 in Section E). Figure 14 in Section E

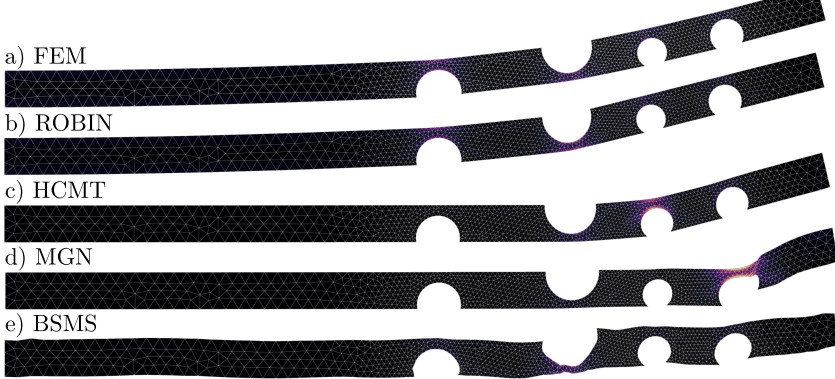

a) FEM

b) ROBIN

c) HCMT

d) MGN

e) BSMS

Figure 5: Comparison of the predicted rollout deformations and von Mises stresses (color, yellow is high) on BENDINGBEAM between **a)** the FEM, **b)** ROBIN, **c)** HCMT, **d)** MGN, and **e)** BSMS. ROBIN is able to accurately reproduce the FEM results. Neither HCMT, MGN nor BSMS are able to resolve global deformation modes, illustrating the importance of the AMPN for global message propagation.

visualizes how high frequency errors accumulate over the rollout if we skip the later denosing steps, demonstrating the importance of reducing the high frequency errors for long rollouts despite low displacement RMSE.

**Diffusion truncation.** Figure 6 visualizes displacement RMSE and fine-mesh edge-wise gradient RMSE $||(\mathbf{u}_i - \mathbf{u}_j)||_2/||(\mathbf{x}_i^0 - \mathbf{x}_j^0)||_2$, against ground truth. Early diffusion steps primarily remove low-frequency error (global RMSE drops), whereas later steps reduce high-frequency error (gradient RMSE). Skipping the later denoising steps results in high frequency error accumulation and mesh degradation throughout the rollout (cf. Figure 14 in Section E), as observed in one step models [9]. With $m=1$, ROBIN does not increase the gradient error on BENDINGBEAM. We observe similar results on DEFORMINGPLATE and IMPACTPLATE, as illustrated in Figure 15 in Section E.

**Ablations.** As shown in Figure 7, ablations confirm the importance of each ROBIN component. Reducing the diffusion length $K$ slightly decreases accuracy across all datasets. However, even with $K=5$ diffusion steps, ROBIN remains significantly more accurate than all baselines. Using

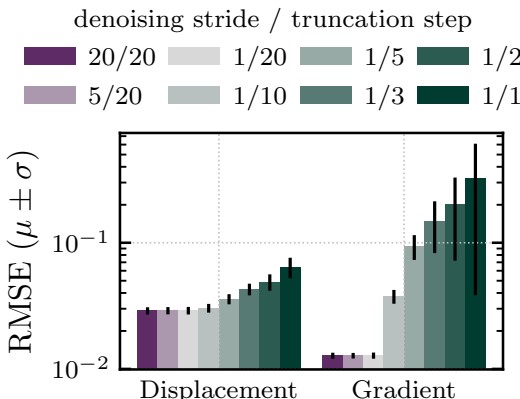

denoising stride / truncation step

- 20/20
- 5/20
- 1/20
- 1/10
- 1/5
- 1/3
- 1/2
- 1/1

Figure 6: Comparison of the predicted rollout displacements and displacement gradients on BENDINGBEAM for different truncation steps $k_{tr}$. While the first diffusion steps strongly decrease the displacement RMSE, the later steps are important to reduce the local displacement gradient RMSE. A fast inference with a low denoising stride neither increases the displacement RMSE nor the gradient RMSE during a rollout.

non-shared layers significantly degrades performance on BENDINGBEAM, likely due to a reduced effective receptive field. The large error increase for *State prediction* indicates that residual prediction is crucial for LAGRANGIAN simulations. Across all datasets, ROBIN outperforms the non-hierarchical variant, the non-diffusion variant, and the combination of DDPMs with the strongest hierarchical baseline (HCMT), demonstrating the synergy between DDPMs and AMPNs.

**Generalization to large meshes.** Fine-tuning the pre-trained ROBIN converges markedly faster and attains substantially lower RMSE than training from scratch, demonstrating mesh-size independence and efficient transfer to deeper AMG hierarchies with twelve times more nodes, without architectural changes. In Figure 8, pretrained predictions are visually indistinguishable from FEM even in thin, high-stress regions with large bending. The model trained from scratch exhibits negligible deformation due to much slower convergence within the same training budget. ROBIN consistently requires 31 s per case, while the numerical solver averages 108 s and can require up to 4248 s, due to problem-dependent nonlinear convergence costs. Table 3 in Section E lists the quantitative results.

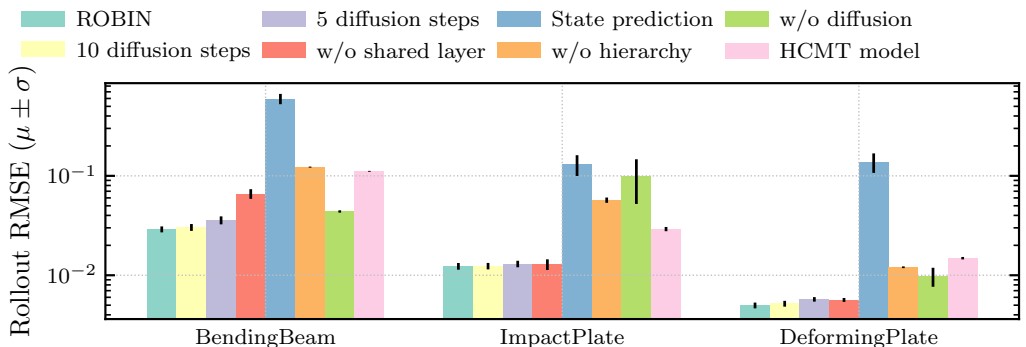

Figure 7: Rollout RMSE of the displacement predictions. ROBIN remains accurate, even for a very small number of diffusion steps. Shared layers are crucial for BENDINGBEAM to increase the receptive field. Replacing residual predictions with state predictions, hierarchical architectures with non-hierarchical architectures or with HCMT architectures, and diffusion with non-diffusion architectures significantly decrease the accuracy across all datasets.

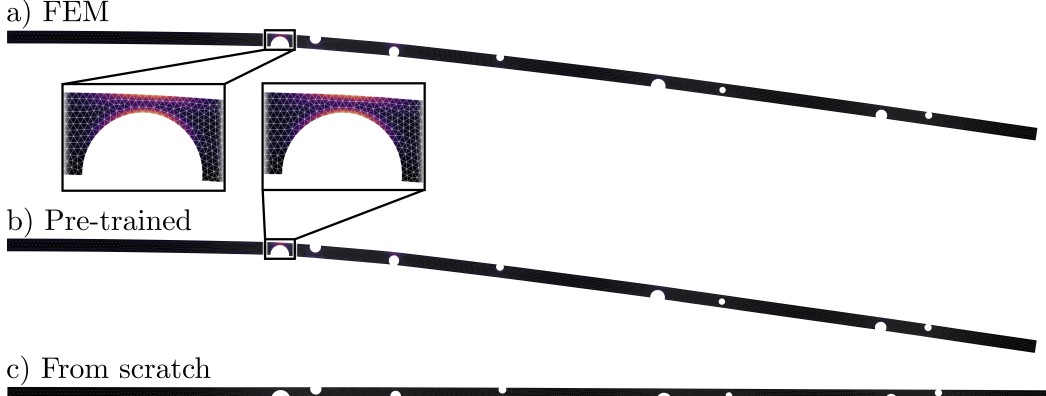

Figure 8: Comparison of the predicted rollout deformations and von Mises stresses (color, yellow is large) on BENDINGBEAMLARGE between **a)** the FEM, **b)** a fine-tuned ROBIN model, which was pre-trained on the small BENDINGBEAM dataset, and, **c)** a ROBIN model trained from scratch for the same number of training iterations. The fine-tuned ROBIN closely matches the FEM deformation and stress distribution, capturing even local high-stress hotspots, whereas the from-scratch model underestimates deformation due to much slower convergence within the same training budget.

## 6 Conclusion

We introduced Rolling Diffusion-Batched Inference Network (ROBIN), a diffusion-based HGNN that utilizes AMPNs to refine mesh-based predictions across scales. Leveraging the expressiveness of multiscale message passing and the accuracy of diffusion, ROBIN outperforms state-of-the-art simulators on varied nonlinear solid mechanics tasks in terms of predictive accuracy. These tasks include a novel BENDINGBEAM dataset that reveals limitations of current learned simulators. ROBI, ROBIN's inference scheme, parallelizes diffusion across time steps, reducing inference runtime by up to an order of magnitude without sacrificing accuracy. We validated ROBIN on three challenging datasets, including the new BENDINGBEAM benchmark, and demonstrated significant gains in accuracy and efficiency. We discuss the broader impact of this work and our method in Appendix A.

**Limitations and Future Work.** ROBIN currently does not possess SO(3) equivariance. Adding this property could improve the accuracy of orientation-sensitive predictions. We focus on DDPM, while other diffusion formulations or denoising schedules could provide valuable insights and further enhance performance. Similarly, our experiments cover nonlinear solid mechanics, and extensions to other domains, such as fluid dynamics, are a promising direction. Lastly, ROBIN's combination of fast inference and high accuracy opens opportunities for accelerating multi-stage design and optimization workflows.

## Acknowledgements

This work is part of the DFG AI Research Unit 5339 regarding the combination of physics-based simulation with AI-based methodologies for the fast maturation of manufacturing processes. The financial support by German Research Foundation (DFG, Deutsche Forschungsgemeinschaft) is gratefully acknowledged. The authors acknowledge support by the state of Baden-Württemberg through bwHPC, as well as the HoreKa supercomputer funded by the Ministry of Science, Research and the Arts Baden-Württemberg and by the German Federal Ministry of Education and Research. This work is supported by the Helmholtz Association Initiative and Networking Fund on the HAICORE@KIT partition.

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

## A  Broader Impact

The ML-based simulator, Rolling Diffusion-Batched Inference Network (ROBIN), offers significant advantages for computational modeling and simulation. This is achieved by reducing computational costs while maintaining accuracy. This enables engineers to iterate through significantly more design variations or to quickly evaluate numerous scenarios using the fast model. However, like all powerful computational tools, there is a risk of misuse, for instance, in weapons development or unsustainable resource exploitation.

## B  Datasets

Table 1 provides an overview of the considered datasets in this work.

Table 1: Comparison of the datasets BENDINGBEAM, BENDINGBEAMLARGE, IMPACTPLATE [22] and DEFORMINGPLATE [5] considered in this work. The column *Nonlinearity* distinguishes three different types: geometry (Geo), material (Mat), and boundary conditions (BC).

| Datasets | Dynamic | Nonlinearity | avg. # Nodes | Mesh Type | Steps T | Dim |
|---|---|---|---|---|---|---|
| BENDINGBEAM | Quasi-Static | Geo | 744 | Triangles | 400 | 2D |
| BENDINGBEAMLARGE | Quasi-Static | Geo | 8897 | Triangles | 100 | 2D |
| IMPACTPLATE | Dynamic | Geo, BC | 2208 | Triangles | 52 | 2D |
| DEFORMINGPLATE | Quasi-Static | Geo, BC, Mat | 1271 | Tetrahedrons | 400 | 3D |

**BENDINGBEAM.** This dataset considers the bending of beam parts due to external forces. Bending is one of the most basic deformation modes of parts in structural mechanics. The dataset is designed as a diagnostic benchmark for neural PDE solvers, addressing various potential bottlenecks. The force and handle boundary conditions are very local, only being defined on a small subset of mesh nodes (cf. Figure 9). However, the resulting deformations affect all nodes.

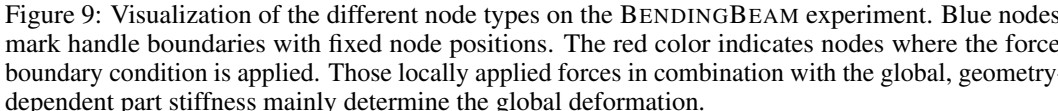

Figure 9: Visualization of the different node types on the BENDINGBEAM experiment. Blue nodes mark handle boundaries with fixed node positions. The red color indicates nodes where the force boundary condition is applied. Those locally applied forces in combination with the global, geometry-dependent part stiffness mainly determine the global deformation.

Hence, the neural PDE must effectively propagate local information to all nodes of the mesh. Next, the dataset considers beams with large aspect ratios. This results in large graph diameters, which represent the shortest path between the most distant nodes. The mesh resolution is increased at thin walls, which additionally increases the graph diameter. The local boundary conditions have to be transmitted across a large number of nodes, which challenges the ability to propagate messages globally. The geometry and especially the thin locations of the geometry strongly influence the global bending stiffness and deformation of the part. Overall, the model has to output accurate solutions across various spatial frequencies.

The solutions are created with scikit-fem [61], iteratively solved using Newton-Raphson until the residual fell below a tolerance of $10^{-8}$. Each simulation is solved for a total number of 400 time steps. We create a total number of 1000 simulations for training, 100 for validation and 100 for testing.

To evaluate the generalization capability of ROBIN, we create an additional dataset variant BENDINGBEAMLARGE, containing meshes more than ten times larger than those in BENDINGBEAM, with an average of 8897 nodes. One simulation in BENDINGBEAMLARGE contains 100 time steps.

## C  Setup

**Hardware and Compute.** We train all models on a single NVIDIA A100 GPU with a maximum training time of 48 hours, while most models required approximately 40 hours. In total, we trained

11 models across 3 datasets, each on 5 seeds: ROBIN, MGN, HCMT, BSMS, as well as 7 ablations of ROBIN. That amounts to $40$ hours$\times 5 \times 3 \times 11 = 6600$ hours training time. Furthermore, we trained two models on BENDINGBEAMLARGE: a pre-trained ROBIN model and a model from scratch, each on 5 seeds for approximately 40 hours, yielding a total duration of $40$ hours $\times 5 \times 2 = 400$ hours. For each training, we required a comparable amount of time for development and hyperparameter tuning. Additionally, we conducted inference experiments to measure the inference speed for 4 models (ROBIN, MGN, HCMT, and BSMS) and 7 ROBIN variants across 5 different seeds. We run inference experiments on 3 datasets, with each experiment taking about 1 hour on average. In total we obtain a runtime of $1$ hour $\times 5 \times 3 \times 11 = 165$ hours for the inference experiments.

**Training.** We implement ROBIN in *PyTorch* [62] and train it with ADAM [63]. We use an exponential learning rate decay, which decreases the learning rate from $1e-4$ to $1e-6$ over the training time, including 1000 linearly increasing warm-up steps. We clip gradients such that their $L_2$-norm doesn't exceed 1. We train ROBIN in BENDINGBEAM with 9M samples and in IMPACTPLATE with 6M samples both with a batch size of 16, resulting in 562,500 and 375,000 training iterations. In DEFORMINGPLATE we reduce the batch size to 12 and train for 300,000 iterations with 3.6M samples.

**Features.** Table 2 provides an overview of the used input and output features for ROBIN. In addition to the default features, we extend the node embeddings of BENDINGBEAM with the force residual $\Delta f_{\text{BC}}$, which is defined by the boundary condition. In IMPACTPLATE, we add the density $\rho_i$ and the Young's modulus $Y_i$ as node features. In DEFORMINGPLATE, we add the scripted displacement residual $\Delta \mathbf{u}_{\text{BC}}$ of the actuator. We normalize all input features based on the training dataset, setting them to have a zero mean and unit variance. We add a small amount of training noise [4, 5] of $10^{-5} \, \sigma_{\mathbf{x}}$ to the node positions $\mathbf{x}_i^t$, where we scale the noise level with the standard deviation of the features $\sigma_{\mathbf{x}}$. For IMPACTPLATE we noise the input history $\Delta \mathbf{u}_{i,0}^{t-1} = \mathbf{x}_i^t - \mathbf{x}_i^{t-1}$ with $10^{-3} \, \sigma_{\mathbf{x}}$ to prevent overfitting on the history.

Table 2: Node $\mathbf{k}_i$ and edge embeddings $\mathbf{e}_{ij}$ for the different datasets, depending on the node $\mathcal{V}$ and edge sets $\mathcal{E}$.

| Datasets | Inputs $\mathcal{V}^0$ | Inputs $\mathcal{V}^{1:L}$ | Inputs $\mathcal{E}^{0:L,\text{M}}$ | Inputs $\mathcal{E}^{0:L,\text{C}}$ | Inputs $\mathcal{E}^{0:L,\text{U/D}}$ | Outputs $\mathcal{V}^{0,\text{M}}$ |
|---|---|---|---|---|---|---|
| BENDINGBEAM | $\mathbf{n}_i, \Delta \mathbf{u}_{i,k}^t, \Delta f_{\text{BC}}$ | $\mathbf{n}_i$ | $\mathbf{x}_{ij}^t, \lvert \mathbf{x}_{ij}^t \rvert, \mathbf{x}_{ij}^0, \lvert \mathbf{x}_{ij}^0 \rvert$ | $\mathbf{x}_{ij}^t, \lvert \mathbf{x}_{ij}^t \rvert$ | $\mathbf{x}_{ij}^t, \lvert \mathbf{x}_{ij}^t \rvert, \mathbf{x}_{ij}^0, \lvert \mathbf{x}_{ij}^0 \rvert$ | $\mathbf{v}_{i,\theta}(\Delta \mathbf{u}_{i,k}^t)$ |
| IMPACTPLATE | $\mathbf{n}_i, \Delta \mathbf{u}_{i,k}^t, \Delta \mathbf{u}_{i,0}^{t-1}, \rho_i, Y_i$ | $\mathbf{n}_i$ | $\mathbf{x}_{ij}^t, \lvert \mathbf{x}_{ij}^t \rvert, \mathbf{x}_{ij}^0, \lvert \mathbf{x}_{ij}^0 \rvert$ | $\mathbf{x}_{ij}^t, \lvert \mathbf{x}_{ij}^t \rvert$ | $\mathbf{x}_{ij}^t, \lvert \mathbf{x}_{ij}^t \rvert, \mathbf{x}_{ij}^0, \lvert \mathbf{x}_{ij}^0 \rvert$ | $\mathbf{v}_{i,\theta}(\Delta \mathbf{u}_{i,k}^t)$ |
| DEFORMINGPLATE | $\mathbf{n}_i, \Delta \mathbf{u}_{i,k}^t, \Delta \mathbf{u}_{\text{BC}}$ | $\mathbf{n}_i$ | $\mathbf{x}_{ij}^t, \lvert \mathbf{x}_{ij}^t \rvert, \mathbf{x}_{ij}^0, \lvert \mathbf{x}_{ij}^0 \rvert$ | $\mathbf{x}_{ij}^t, \lvert \mathbf{x}_{ij}^t \rvert$ | $\mathbf{x}_{ij}^t, \lvert \mathbf{x}_{ij}^t \rvert, \mathbf{x}_{ij}^0, \lvert \mathbf{x}_{ij}^0 \rvert$ | $\mathbf{v}_{i,\theta}(\Delta \mathbf{u}_{i,k}^t)$ |

**Hierarchical Graph.** Since the relative motion of the components in the considered experiments is not too large, we define the contact edges based on the initial mesh configuration and keep them fixed to maintain a constant graph. In DEFORMINGPLATE we set the contact radius to $R = 0.1$, connecting actuator nodes with plate nodes. In IMPACTPLATE we connect ball nodes and plate nodes with a radius of $R = 1.2$. In all three experiments, we create $L = 2$ coarse layers to obtain 3 mesh levels.

**Algebraic-hierarchical Message Passing Networks.** We use 3 Pre- and 3 Post-processing layers, 2 Up- and 2 Downsampling layers and 5 Solving layers, which yields a total number of 15 learnable layers. We add a layer norm before each MLP and use two linear layers, a hidden size of 128 and a Sigmoid Linear Unit (SiLU) [64] activation function. A max aggregation is used in all message passing layers.

**Denoising Diffusion Probabilistic Models.** We use $K = 20$ denoising steps and a denoising stride of $m = 5$ for ROBIN by default. The $\beta$ variances of the DDPM scheduler are geometrically spaced for training and inference, starting from a minimum noise variance of $1e-4$ (for $\beta_1$) and going up to $1.0$ (for $\beta_K$).

**Metrics.** To compare the rollout accuracy, we follow [22] and define the Root Mean Squared Euclidean distance error $\text{RMSE} = \sqrt{1/(N_i N_j) \sum_{i=1}^{N_i} \sum_{j=1}^{N_j} (\tilde{u}_{ij} - u_{ij})^2}$, where the prediction $\tilde{u}_{ij}$

and the ground truth $u_{ij}$ have $N_i$ nodes and $N_j$ features. We then calculate the mean over all time steps, the mean over the dataset and finally the mean $\mu$ and standard deviation $\sigma$ over the 5 seeds.

## D  Baselines, Ablations and Variants

**Baselines.** We use the official *TensorFlow* [65] implementation of the authors for the baselines HCMT[3] [22] and MGN[4] [5]. We use ADAM [63] for training HCMT and MGN with an exponential learning rate decay from $1e-4$ to $1e-6$, a batch size of 1 and a hidden size of 128. We use 15 message-passing steps for MGN on all datasets, as well as a total of 15 Hierarchical Mesh Transformer (HMT) and Contact Mesh Transformer (CMT) layers for HCMT.

On BENDINGBEAM, we train HCMT for 4M training iterations with a training noise [4, 5] of 0.001. We maximize the receptive field and set the number of mesh levels to 5, the maximum at which at least five nodes remain available across all meshes in the dataset, required for Delaunay remeshing [66]. This results in 9 HMTs. Since BENDINGBEAM is a contact-free task, we replaced the dual-branch CMT by 6 single-branch Mesh Transformer layers that only attend to mesh edges instead of mesh and contact edges. We use the same architecture and hyperparameter for HCMT on IMPACTPLATE and DEFORMINGPLATE as proposed by the authors [22], and train it for 3M steps and 2M steps, respectively.

We follow the authors' implementation and add world edges to the mesh graph of MGN instead of contact edges to increase the receptive field of the non-hierarchical architecture. MGN is trained for 3M iterations on BENDINGBEAM and uses a training noise of 0.001 with a world edge radius of $R = 0.13$. On IMPACTPLATE we train MGN for 3M steps and use a world edge radius of $R = 0.03$ and a training noise of 0.003. We train MGN on DEFORMINGPLATE for 1.5M steps and use the authors' proposed settings [5]. To prevent out-of-memory errors in edge cases on DEFORMINGPLATE, we restrict the number of world edges to 200,000 by selecting those with the smallest node distances.

For BSMS [48], we use the official *PyTorch* [62] implementation[5] of the authors. We follow the authors and use the maximum number of hierarchy levels possible to maximize the receptive field. We train BSMS on BENDINGBEAM with a batch size of 12 and 6M training samples, i.e., 500K training iterations. We use a training noise of 0.001 and 5 mesh levels. On IMPACTPLATE we train BSMS with a batch size of 8 and 5M samples (625K training iterations), a training noise of 0.003, a contact radius of $R = 0.4$, and 7 hierarchy levels. We train BSMS on DEFORMINGPLATE with a batch size of 8 and 6M samples (750K training iterations), a training noise of 0.003, a contact radius of $R = 0.03$, and 6 hierarchy levels.

**Ablations.** The *10 diffusion steps* and *5 diffusion steps* ablation use the same settings as ROBIN, despite the reduced number of diffusion steps $K$. For the *w/o hierarchy* ablation we use the fine mesh graph $\mathcal{G}^0$ instead of the hierarchical graph $\mathcal{G}^{0:L}$, replace our AMPN by a single Intra-MP-Stack with 15 learnable message passing steps and remove the positional level encoding. In addition, we follow MGN and replace contact edges with world edges to increase the receptive field. To stay within the training budget, we reduce the number of training samples to 1.2M for BENDINGBEAM and to 8M for IMPACTPLATE. For DEFORMINGPLATE we reduce the batch size to 1 and the training samples to 0.6M and also restrict the number of world edges to 200,000 as for MGN. The *w/o diffusion* ablation trains the AMPN with an MSE loss to predict directly the displacement residual $\Delta\mathbf{u}_{i,0}^t$ and uses a one-step autoregressive rollout, such as HCMT, MGN, and BSMS. We use the same training noise settings as the baselines to stabilize the rollouts. The *w/o shared layer* ablation uses a total number of 15 non-shared learnable message passing layers distributed as follows: 1 Pre-Processing and 1 Post-Processing layer per level, 1 Up- and 1 Downsampling layer between each level, and 5 Solving layers. The faster predictions allow an increase in the number of training samples to 11M for BENDINGBEAM, to 8M for IMPACTPLATE, and to 4.6M for DEFORMINGPLATE. For the *HCMT model* ablation, we replace the AMPNs with HCMTs. More specifically, we use the same mesh hierarchy and the same model architecture as HCMTs. Everything else remains the same in ROBIN, including DDPMs and ROBI.

---

[3]`https://github.com/yuyudeep/hcmt/tree/main`
[4]`https://github.com/google-deepmind/deepmind-research/tree/master/meshgraphnets`
[5]`https://github.com/Eydcao/BSMS-GNN/tree/main`

# E    Results

**AMG-based mesh coarsening.**    Root-node AMG coarsening [53] preserves mesh geometry and connectivity (cf. Figure 10). The coarse mesh remains well-aligned with thin geometrical features, and smoothed transfer operators yield wider, algebraically informed receptive fields compared to bi-stride pooling. This fidelity is crucial for predicting geometrically nonlinear deformations.

a) Fine mesh

b) Bi-stride-coarsening and Delaunay-remeshing

c) AMG-based coarsening

d) AMG-based down-/upsampling graphs.

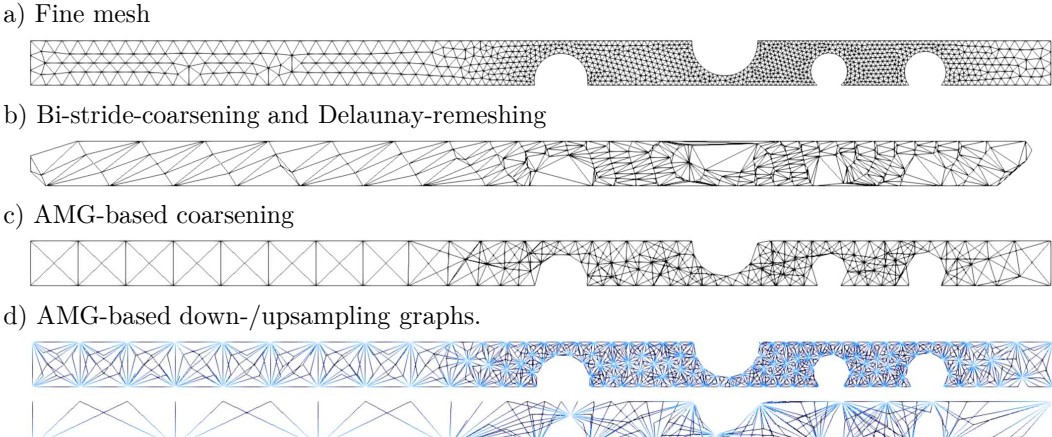

Figure 10: Comparison of AMG-based mesh coarsening to Bi-stride-coarsening and Delaunay-remeshing (BSDL). **a)** the original, fine mesh. **b)** the mesh after two BSDL coarsening steps. **c)** the mesh after one AMG coarsening step. This mesh has approximately as many nodes as the mesh in **b)**. **d)** up- and downsampling edges from level 0 to 1 (top) and level 1 to 2 (bottom). Bright blue indicates coarse nodes of level 1 (top) or 2 (bottom), respectively.

**Inference speed.**    Each point in Figure 11 corresponds to a rollout setting of ROBIN on IMPACT-PLATE and DEFORMINGPLATE. As for BENDINGBEAM in Section 5, decreasing the truncation step reduces wall time while a truncation step of $k_{tr}=2$ is already sufficient to obtain a higher accuracy than all baselines across all datasets (cf. Table 3). The ROBIN default $(1/20)$ is significantly faster than conventional diffusion inference and is even slightly more accurate on IMPACTPLATE, which we attribute to anchoring low-frequency components and reducing drift.

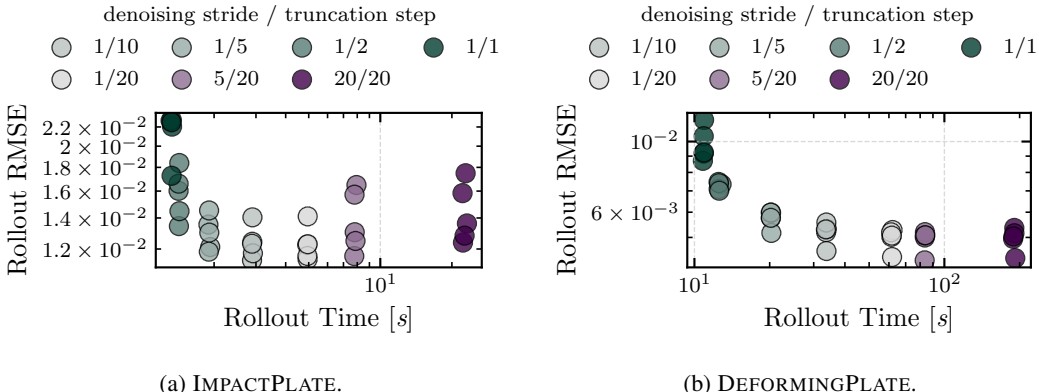

(a) IMPACTPLATE.                                    (b) DEFORMINGPLATE.

Figure 11: RMSE rollout error and inference time of different inference variants of ROBIN on a) IMPACTPLATE and b) DEFORMINGPLATE. ROBIN, i.e., the variant (1/20), is most accurate on IMPACTPLATE and on par on DEFORMINGPLATE with the slower variants (5/20) and conventional inference (20/20). Reducing the truncation step $k_{tr}$ increases speed while decreasing accuracy. The one step variant (1/1) achieves the largest speed-up on DEFORMINGPLATE, but also loses the most accuracy there.

**Baselines.** Figure 12 and Figure 13 visualize the rollout displacement and von Mises stress prediction of ROBIN, HCMT, MGN, and BSMS on IMPACTPLATE and DEFORMINGPLATE, respectively.

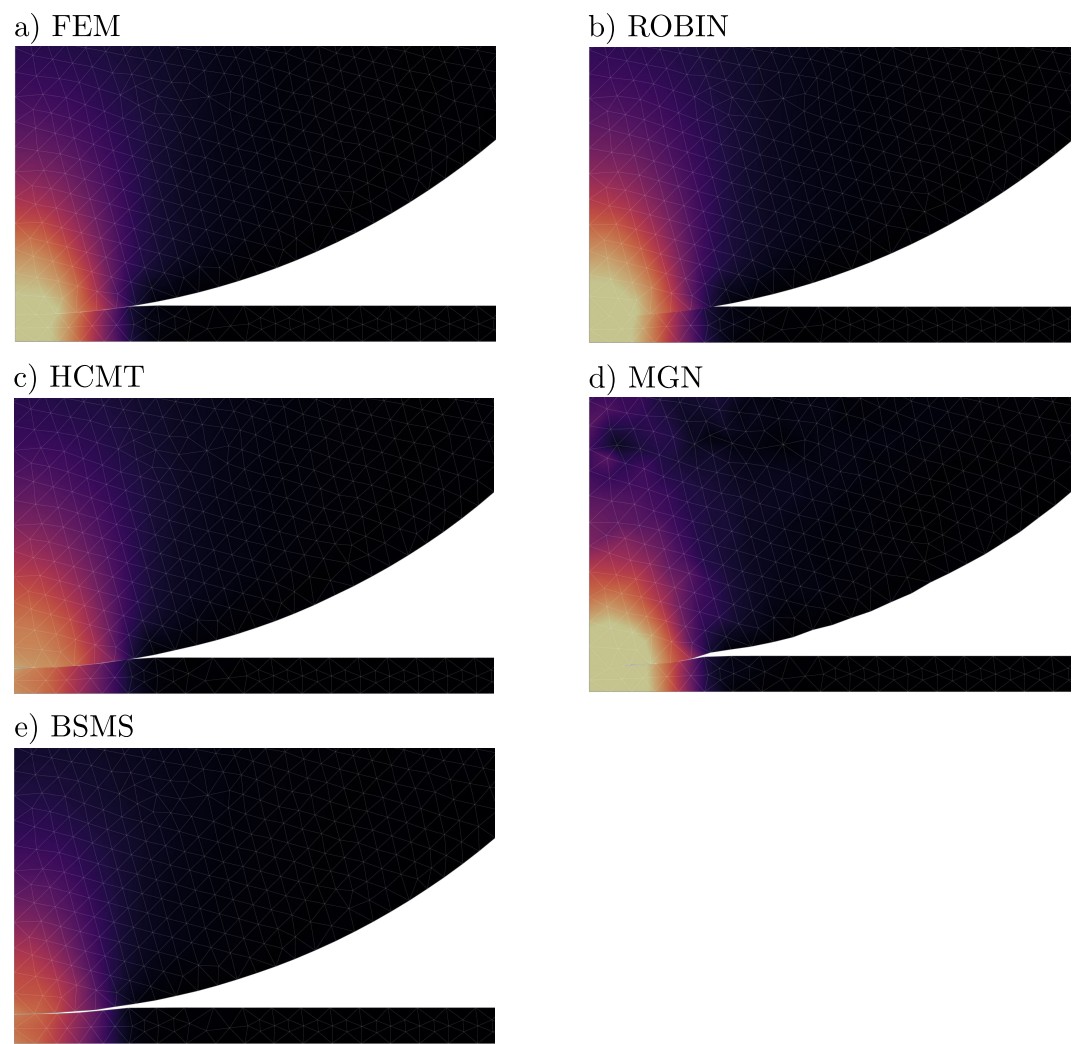

Figure 12: Comparison of the rollout deformation prediction and von Mises stress prediction (color, yellow is high) on IMPACTPLATE to the ground truth of the FEM. ROBIN most accurately resolves the deformation at the contact surface and the resulting stress. The deformation prediction of HCMT and BSMS are close to the FEM prediction, though stress is underestimated. MGN predicts accurately the global modes but exhibits local disturbances.

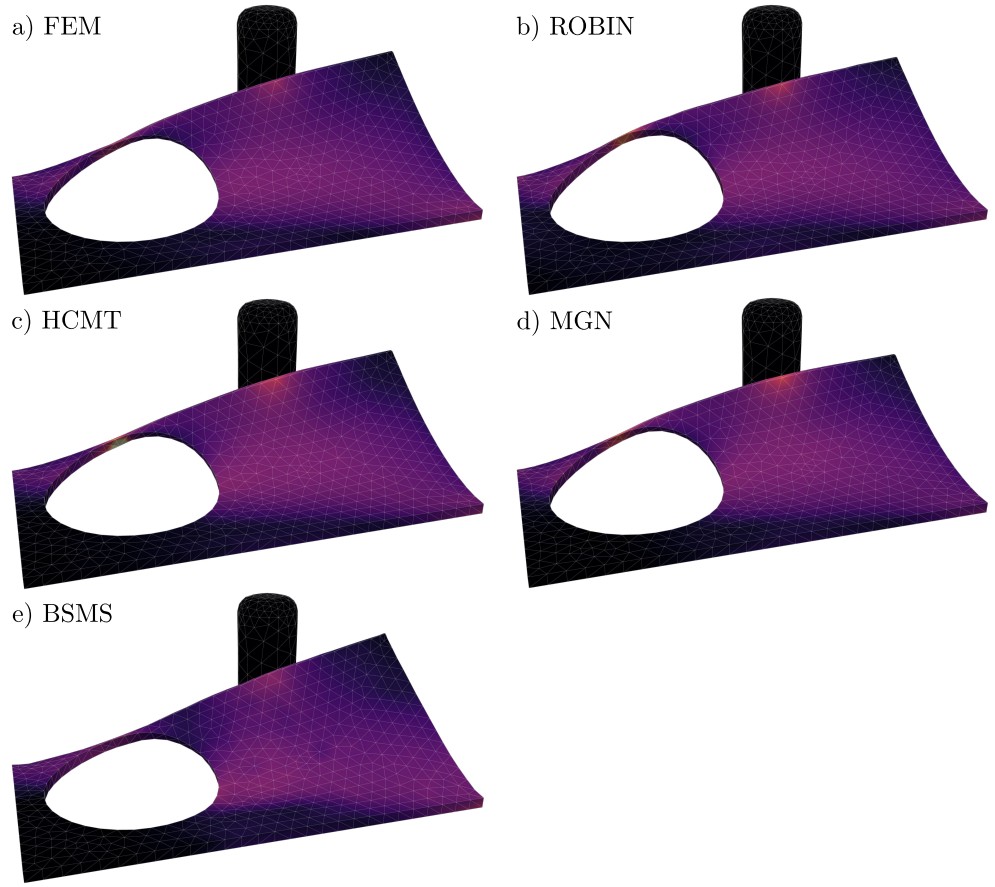

Figure 13: Rollout deformation and von Mises stress prediction (color, yellow is high) on DE-FORMINGPLATE of ROBIN, the baselines and the FEM. All models accurately reproduce the part deformation. HCMT slightly overestimates and BSMS slightly underestimates the stress at the thin wall between the hole and the boundary.

**Diffusion truncation.** The rollouts in Figure 14 across truncation steps $k_{tr}$ illustrate coarse-to-fine frequency behavior of ROBIN. Early steps capture the global, low-frequency shape, whereas additional steps sharpen high-frequency details, e.g., high stresses in thin geometrical features. Even with early truncation, ROBIN maintains coherent global modes. Longer schedules primarily refine local features and minimize the accumulation of high-frequency errors, while retaining the global deformation pattern.

Figure 15 visualize the rollout displacement RMSE and displacement gradient RMSE of different ROBIN variants on DEFORMINGPLATE and IMPACTPLATE. As for BENDINGBEAM, early denoising steps are critical for reducing global displacement error. Later diffusion steps focus on high-frequency solution components. On IMPACTPLATE, the gradient RMSE shows a modest increase while ROBIN attains the lowest displacement RMSE overall. We hypothesize that partially denoising states (small $m$) stabilizes low-frequency components and reduces drift. In contrast, fully denoising (larger $m$) suppresses short-term, high-frequency error accumulation. However, this trade-off does not affect other datasets with significantly longer rollouts.

**Quantitative results.** Table 3 lists the quantitative results of all experiments considered in this work and used for the visualizations.

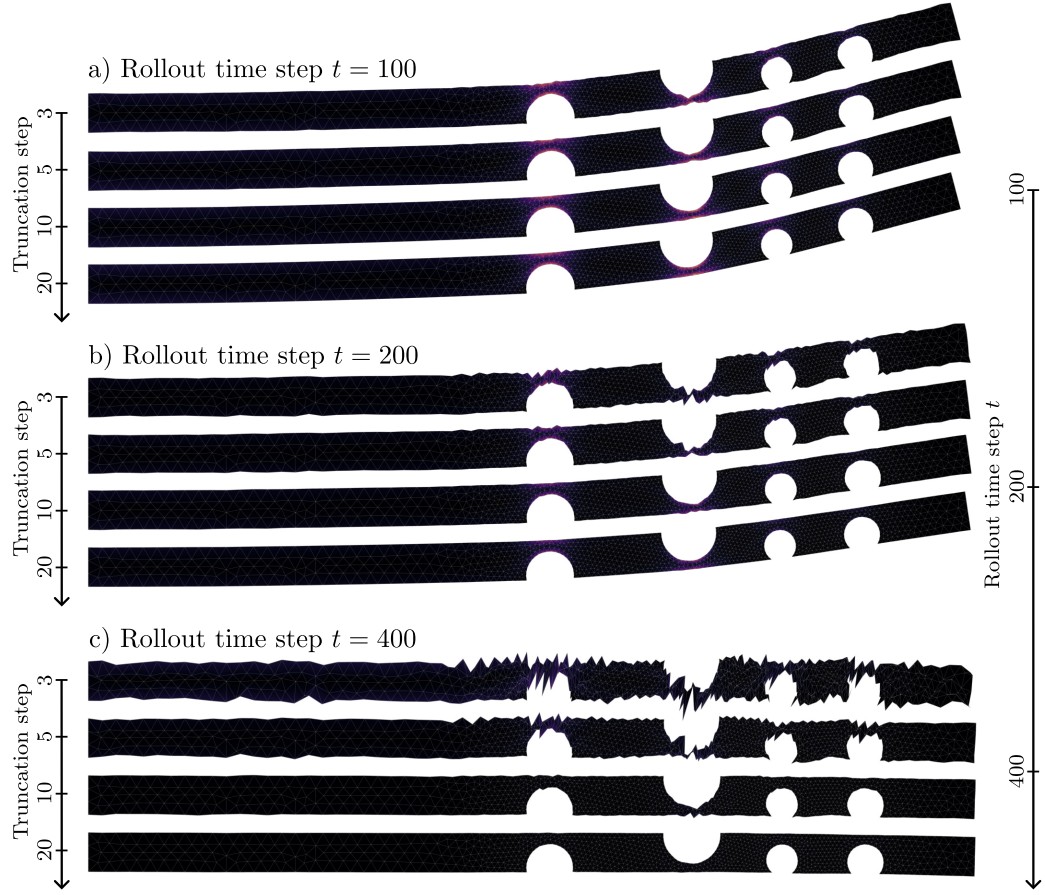

Figure 14: Effect of diffusion truncation on rollouts. Figures a), b), and c) show predicted deformations and von Mises stress prediction (color, yellow is high) at $t = 100$, $t = 200$, and $t = 400$, respectively. In each figure, rows correspond to truncation steps $k_{\text{tr}} = 3, 5, 10, 20$. Using a low truncation step $k_{\text{tr}}$ increases inference speed and enables robust predictions of the global deformation modes. However, it also causes local mesh degradation due to the accumulation of high-frequency errors, as observed in MSE trained one step models [9].

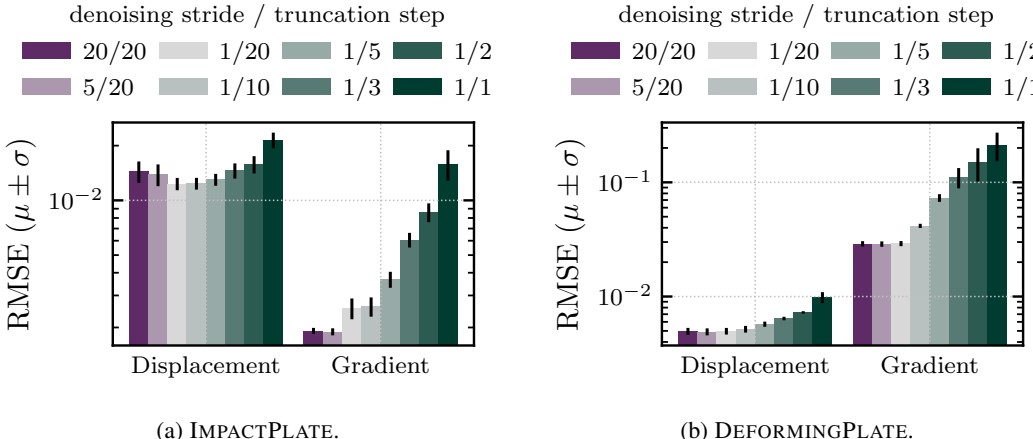

(a) IMPACTPLATE.

(b) DEFORMINGPLATE.

Figure 15: Comparison of the rollout displacement RMSE and displacement gradient RMSE for different ROBIN setting (denoising stride $m$ / truncation step $k_{\text{tr}}$) on a) IMPACTPLATE and b) DEFORMINGPLATE. Early diffusion steps substantially reduce the displacement RMSE and later steps the local displacement gradient RMSE. The fast inference setting (1/20) remains accurate and yields slightly lower displacement RMSE on IMPACTPLATE, but also slightly higher gradient RMSE.

Table 3: Quantitative results on BENDINGBEAM, IMPACTPLATE, DEFORMINGPLATE, and the large-mesh dataset BENDINGBEAMLARGE. We report displacement RMSE $[10^{-3}]$, gradient RMSE of the displacement field $[10^{-3}]$, and rollout wall-clock time [s]. Values are mean $\pm$ std over the test set across 5 seeds (solver timings excluded). The row *Numerical solving* gives the average runtime of the high-fidelity solver (maximum in parentheses). *Baselines* contrasts different models, *Variants* sweep the ROBI settings (denoising stride / truncation step) and additionally report Gradient RMSE, *Ablations* remove ROBIN components, and *Generalization to large meshes* evaluates the upscaling capabilities of ROBIN.

| | BENDINGBEAM | | IMPACTPLATE | | DEFORMINGPLATE | |
|---|---|---|---|---|---|---|
| **Numerical solving** | Time [s] | | Time [s] | | Time [s] | |
| | 46.2 (max. 186.1) | | 742.6 [22] | | 1157.2 [5] | |
| **Baselines** | | | | | | |
| | RMSE $[10^{-3}]$ | Time [s] | RMSE $[10^{-3}]$ | Time [s] | RMSE $[10^{-3}]$ | Time [s] |
| ROBIN | $29.00 \pm 1.00$ | $15.03 \pm 0.04$ | $12.33 \pm 0.96$ | $4.94 \pm 0.01$ | $4.98 \pm 0.33$ | $61.60 \pm 0.29$ |
| HCMT | $121.53 \pm 1.87$ | $25.82 \pm 0.28$ | $19.57 \pm 0.38$ | $4.33 \pm 0.05$ | $8.04 \pm 0.13$ | $31.57 \pm 0.14$ |
| MGN | $189.52 \pm 70.87$ | $21.40 \pm 0.15$ | $54.07 \pm 5.88$ | $2.93 \pm 0.03$ | $8.76 \pm 0.29$ | $24.42 \pm 0.22$ |
| BSMS | $141.98 \pm 7.96$ | $4.70 \pm 0.04$ | $63.52 \pm 32.36$ | $2.16 \pm 0.01$ | $13.59 \pm 5.21$ | $13.28 \pm 0.07$ |
| **Variants (denoising stride / truncation step)** | | | | | | |
| | RMSE $[10^{-3}]$ | Time [s] | RMSE $[10^{-3}]$ | Time [s] | RMSE $[10^{-3}]$ | Time [s] |
| 20/20 | $28.90 \pm 1.91$ | $162.43 \pm 0.89$ | $14.43 \pm 1.93$ | $22.61 \pm 0.36$ | $4.96 \pm 0.36$ | $190.10 \pm 1.33$ |
| 5/20 | $28.99 \pm 1.93$ | $43.39 \pm 0.52$ | $13.87 \pm 1.90$ | $7.83 \pm 0.06$ | $4.92 \pm 0.35$ | $83.61 \pm 0.16$ |
| 1/20 | $29.00 \pm 1.00$ | $15.03 \pm 0.04$ | $12.33 \pm 0.96$ | $4.94 \pm 0.01$ | $4.98 \pm 0.33$ | $61.60 \pm 0.29$ |
| 1/10 | $30.35 \pm 2.41$ | $11.03 \pm 0.10$ | $12.38 \pm 0.92$ | $2.90 \pm 0.01$ | $5.18 \pm 0.35$ | $33.71 \pm 0.10$ |
| 1/5 | $35.81 \pm 3.30$ | $9.88 \pm 0.19$ | $13.01 \pm 0.98$ | $1.91 \pm 0.01$ | $5.73 \pm 0.30$ | $20.24 \pm 0.04$ |
| 1/3 | $42.83 \pm 4.54$ | $9.57 \pm 0.11$ | $14.58 \pm 1.40$ | $1.55 \pm 0.01$ | $6.42 \pm 0.23$ | $14.85 \pm 0.11$ |
| 1/2 | $48.94 \pm 7.25$ | $9.47 \pm 0.11$ | $15.79 \pm 1.71$ | $1.42 \pm 0.00$ | $7.26 \pm 0.18$ | $12.59 \pm 0.13$ |
| 1/1 | $64.23 \pm 11.86$ | $9.49 \pm 0.12$ | $21.46 \pm 2.12$ | $1.32 \pm 0.01$ | $10.79 \pm 0.52$ | $7.11 \pm 0.04$ |
| | Gradient RMSE $[10^{-3}]$ | | Gradient RMSE $[10^{-3}]$ | | Gradient RMSE $[10^{-3}]$ | |
| 20/20 | $12.75 \pm 0.70$ | | $1.92 \pm 0.07$ | | $28.98 \pm 1.56$ | |
| 5/20 | $12.74 \pm 0.65$ | | $1.89 \pm 0.08$ | | $28.83 \pm 1.57$ | |
| 1/20 | $12.77 \pm 0.72$ | | $2.55 \pm 0.33$ | | $29.10 \pm 1.50$ | |
| 1/10 | $37.58 \pm 4.71$ | | $2.61 \pm 0.30$ | | $41.59 \pm 1.76$ | |
| 1/5 | $94.00 \pm 20.90$ | | $3.68 \pm 0.37$ | | $73.13 \pm 5.64$ | |
| 1/3 | $147.97 \pm 65.13$ | | $6.07 \pm 0.56$ | | $110.96 \pm 22.33$ | |
| 1/2 | $200.44 \pm 128.29$ | | $8.61 \pm 1.02$ | | $150.33 \pm 48.48$ | |
| 1/1 | $324.05 \pm 285.65$ | | $23.41 \pm 4.85$ | | $157.30 \pm 8.34$ | |
| **Ablations** | | | | | | |
| | RMSE $[10^{-3}]$ | | RMSE $[10^{-3}]$ | | RMSE $[10^{-3}]$ | |
| ROBIN | $29.00 \pm 1.00$ | | $12.33 \pm 0.96$ | | $4.98 \pm 0.33$ | |
| 10 diff. steps | $33.28 \pm 2.43$ | | $14.58 \pm 1.99$ | | $5.86 \pm 0.61$ | |
| 5 diff. steps | $41.33 \pm 6.27$ | | $15.60 \pm 1.74$ | | $6.40 \pm 0.56$ | |
| w/o shared layer | $66.05 \pm 7.36$ | | $12.88 \pm 1.58$ | | $5.65 \pm 0.26$ | |
| State prediction | $596.38 \pm 70.93$ | | $130.50 \pm 30.80$ | | $137.61 \pm 30.54$ | |
| w/o hierarchy | $122.38 \pm 1.52$ | | $57.02 \pm 3.48$ | | $12.04 \pm 0.24$ | |
| w/o diffusion | $43.83 \pm 1.16$ | | $99.47 \pm 47.30$ | | $9.78 \pm 2.12$ | |
| HCMT model | $111.05 \pm 1.15$ | | $29.18 \pm 1.43$ | | $14.86 \pm 0.42$ | |

| | BENDINGBEAMLARGE | |
|---|---|---|
| **Numerical solving** | Time [s] | |
| | 108.30 (max. 4248.01) | |
| **Generalization to large meshes** | | |
| | RMSE $[10^{-3}]$ | Time [s] |
| Pre-trained | $77.59 \pm 4.97$ | $30.94 \pm 0.63$ |
| From scratch | $215.50 \pm 12.14$ | $30.95 \pm 0.58$ |

