# OpenReview forum: "Diffusion-Based Hierarchical Graph Neural Networks for Simulating Nonlinear Solid Mechanics"
_NeurIPS.cc/2025/Conference — NeurIPS 2025 spotlight_

### Official Review · Reviewer_V9Hd · 2025-06-10

**Clarity:** 3
**Significance:** 2
**Originality:** 2
**Rating:** 5
**Confidence:** 4

**Summary:**

The paper introduces a method for learned, GNN-based simulation targeted at solid mechanics applications. It introduces a novel rolling diffusion method and hierarchical message passing scheme which allows fast & accurate predictions.

**Questions:**

No major open questions. Overall I think this a paper worth accepting as this is an interesting approach which shows progress on a challenging problem. I would however encourage the authors to think about more thoroughly investigating the effect of some of the choices made in the paper, either as follow-up work if this paper is accepted, or a major revision of this paper if it is not.
E.g. I'd love to see a proper review of different ways to perform HGNN, and how this affects different physical domains, from smooth simulations with long-range effects such as the ones studied here, to local chaotic CFD simulations. Similarly, a deep dive into the various choices in diffusion & denoising and the effect this has on consistency and frequency distribution could be very interesting.

**Ethical Concerns:**

["NO or VERY MINOR ethics concerns only"]

**Final Justification:**

I think this is a promising paper, and I'm in favor of accepting it to NeurIPS. The authors addresses all the issues raised in review, and performed additional experiments which will further strengthen the paper.

**Limitations:**

yes, except for the points mentioned above.

**Quality:**

3

**Strengths And Weaknesses:**

Strengths:
- Quasistatic elasticity simulations are challenging for GNN simulators; yet the presented approach does seem to be quite effective at them
- All the absolutely necessary ablations are there (even though comparisons to other HGNN or diffusion approaches would have been nice)
- This is a nice demonstration that rolling diffusion can be a useful approach for learned simulators
- Overall, the paper is well written and easy to follow

Weaknesses:
- The main issue I see is that the paper splits its attention too much between rolling diffusion, HGNNs and applications in solid mechanics. The paper does the minimum diligence in all of these areas, but doesn't deliver any fundamental insights in either that might transfer to other work.
E.g., there are so many HGNN papers with slightly different choices of graph construction and message propagation, and this paper adds yet another option without properly evaluating it against other options. Similarly, I feel the effect of conditioning on not fully denoised previous timesteps is underexplored. There is one ablation on this with downstream performance, but it would have been much more interesting to develop an understanding to what this does to simulation rollouts; e.g. does this affect coherence of high-frequency details?
- The choice of experimental domain is a bit odd. Quasistatic elasticity is a good domain for studying hierarchical message passing, but less so for diffusion-- these simulations are very deterministic and smooth. So diffusion a) seems much less important here but also b) the tradeoffs in conditioning on not fully resolved diffusion might look very different in a less smooth domain. So it would have been good to also include a domain with more chaotic dynamics on multiple scales (e.g. CFD, weather, fracture,...).
- (Minor) The description of ROBI and corresponding figure could be improved; it's a bit of a notation-fest, and the figure is hard to parse. It took me a few passes to understand which step is conditioned on what exactly in diffusion & simulation time.

---

> ### Author Rebuttal · Authors · 2025-07-31
>
> We appreciate the positive comments of the author on the effectiveness of our approach for quasistatic elasticity, the inclusion of necessary ablations and the clarity of the paper. We address the concerns raised below.
>
> > The choice of experimental domain is a bit odd. Quasistatic elasticity is a good domain for studying hierarchical message passing, but less so for diffusion-- these simulations are very deterministic and smooth.
> >
>
> It might seem counterintuitive, but we found that preserving smoothness is often difficult for HGNNs, which makes diffusion attractive in this domain. Models trained on an MSE loss must accurately capture global deformations, and even slight shifts or rotations can increase error. HGNNs often accumulate errors from high frequencies that are less penalized, which leads to local mesh distortions and drifting out of the training distribution over time. Non-smooth solutions result in significant deviations in the displacement gradient, which directly impacts stress.
>
> To analyze this behavior, we added a truncation parameter to the ROBIN algorithm. This allows us to stop the denoising process early. For example, we can stop after the first $K_I = 5$ diffusion steps (instead of denoising all $K = 20$ steps) and treat the reconstructed clean sample as our ground truth. Regarding the rollout RMSE, we have evaluated ROBIN for different truncation valuess $K_I$. For each edge $(i,j)$ of the fine mesh, we have also evaluated the gradient  $||(\\mathbf{u}\_{i} - \\mathbf{u}\_{j})||_2 / ||(\\mathbf{x}\_{i} - \\mathbf{x}\_{j})||_2$ along that edge and calculated the RMSE.
>
> |  |  | **BendingBeam** |  | **DeformingPlate** |  | **ImpactPlate** |  |
> | --- | --- | --- | --- | --- | --- | --- | --- |
> |  | $K_I$ | Rollout RMSE [$10^{-3}$] | Rollout gradient RMSE [$10^{-3}$] | Rollout RMSE [$10^{-3}$] | Rollout gradient RMSE [$10^{-3}$] | Rollout RMSE [$10^{-3}$] | Rollout gradient RMSE [$10^{-3}$] |
> | ROBIN - one step | 1 | 64.232 +/- 11.862 | 324.047 +/- 285.654 | 10.787 +/- 0.521 | 157.279 +/- 8.338 | 21.462 +/- 2.115 | 23.414 +/- 4.853 |
> | ROBIN - two step | 2 | 48.936 +/- 7.248  | 200.437 +/- 128.285 | 7.263 +/- 0.179 | 150.325 +/- 48.476 | 15.788 +/- 1.714 | 8.609 +/- 1.019 |
> | ROBIN - three step | 3 | 42.831 +/- 4.536 | 147.966 +/- 65.129 | 6.422 +/- 0.225 | 110.958 +/- 22.329 | 14.576 +/- 1.396 | 6.066 +/- 0.560 |
> | ROBIN - five step | 5 | 35.812 +/- 3.299 | 93.999 +/-  20.897 | 5.729 +/- 0.299 | 73.134 +/- 5.636 | 13.011 +/- 0.977 | 3.677 +/- 0.371 |
> | ROBIN - ten step | 10 | 30.349 +/- 2.409 | 37.582 +/- 4.711 | 5.181 +/- 0.347 | 41.587 +/- 1.758 | 12.379 +/- 0.923 | 2.610 +/- 0.316 |
> | ROBIN | 20 | 29.002 +/- 1.997 | 12.765 +/- 0.716 | 4.960 +/- 0.355 | 29.098 +/- 1.494 | 12.330 +/- 0.958 | 2.553 +/- 0.334 |
>
> The initial diffusion steps concentrate on global solution frequencies, significantly reducing the global RMSE. Later steps focus on high frequencies, reducing local gradient error. Over time, these high-frequency errors can lead to mesh distortions, as observed in MSE-trained HGNNs.
>
> > Similarly, I feel the effect of conditioning on not fully denoised previous timesteps is underexplored. […] e.g. does this affect coherence of high-frequency details? […] Similarly, a deep dive into the various choices in diffusion & enoising and the effect this has on consistency and frequency distribution could be very interesting.
> >
>
> We agree that investigating how conditioning partially denoised states affects the high-frequency component would be helpful. The table below shows how the denoising stride $m$ affects the rollout RMSE and rollout gradient RMSE for all the datasets.
>
> |  |  | **BendingBeam** |  | **DeformingPlate** |  | **ImpactPlate** |  |
> | --- | --- | --- | --- | --- | --- | --- | --- |
> | Inference variants | $m$ | Rollout RMSE [$10^{-3}$] | Rollout gradient RMSE [$10^{-3}$] | Rollout RMSE [$10^{-3}$] | Rollout gradient RMSE [$10^{-3}$] | Rollout RMSE [$10^{-3}$] | Rollout gradient RMSE [$10^{-3}$] |
> | ROBIN | 1 | 29.002 +/- 1.997 | 12.765 +/- 0.716 | 4.960 +/- 0.355 | 29.098 +/- 1.494 | 12.330 +/- 0.958 | 2.553 +/- 0.334 |
> | Modest denoising stride | 5 | 28.987 +/- 1.932 | 12.741 +/- 0.653 | 4.916 +/- 0.346 | 28.828 +/- 1.566 | 13.865 +/- 1.900 | 1.894 +/- 0.084 |
> | Conventional Inference | 20 | 28.896 +/- 1.906 | 12.747 +/- 0.696 | 4.960 +/- 0.355 | 28.977 +/- 1.556 | 14.431 +/- 1.933 | 1.916 +/- 0.074 |
>
> ROBIN with $m=1$, in which the previous physical time step is always denoised only one diffusion step further,  does not increase gradient error on BendingBeam or DeformingPlate. However, there is a modest increase on ImpactPlate, and ROBIN achieves the lowest RMSE overall. We hypothesize that partially denoising states keeps low-frequency components anchored and reduces drift. In contrast, fully denoising states  (larger $m$) reduces short-term, high-frequency error accumulation. However, this phenomenon does not seem to affect long rollouts. We plan to investigate this further in future work, as well as how it is influenced by alternative diffusion backbones and schedulers.
>
> > […] there are so many HGNN papers with slightly different choices of graph construction and message propagation, and this paper adds yet another option without properly evaluating it against other options.
> >
>
> We acknowledge the many HGNNs in the literature and emphasize that our primary contribution is the synergy of diffusion with our AMPNs, not a new HGNN. Our AMPNs follow the architecture of algebraic multigrid methods to solve for frequencies at different scales. For example, we use an explicit solver layer at the coarse level to solve for global frequencies and shared intra-level message passing stacks for high frequencies w.r.t. the mesh scale of the level. We combine this approach with diffusion to specifically address all frequencies during training and inference. Our ablation studies highlight the synergy between the two. We lose accuracy on all tasks if we remove one of them, but we significantly outperform the baselines when they are used together.
>
> To further demonstrate the benefits of combining our AMPNs with diffusion, we conducted an ablation study on the Bending Beam dataset, replacing our AMPNs with the HCMT architecture. Additionally, we compared our results to those of another hierarchical GNN baseline: the Bi-stride multiscale-Graph Neural Networks (BSMS-GNNs) [1].
>
> |  | **BendingBeam** |
> | --- | --- |
> |  | Rollout RMSE [$10^{-3}$] |
> | ROBIN | 28.896 +/- 1.906 |
> | ROBIN - HCMT | 111.049 +/- 1.146 |
> | HCMT | 121.526 +/- 1.865 |
> | BSMS-GNNs | 141.979 +/- 7.960 |
> | MGN | 189.515 +/- 70.865 |
>
> Although integrating diffusion into HCMT improves RMSE compared to HCMT alone, ROBIN significantly outperforms both. BSMS-GNNs are nearly as accurate as the HCMT baseline, but significantly more accurate than non-hierarchical MeshGraphNets. Only the combination of diffusion and AMPNs can accurately resolve the wide frequency spectrum of BendingBeam.
>
> To further demonstrate the advantages of the AMPN architecture, we have fine-tuned ROBIN on the BendingBeamLarge dataset. This dataset contains simulations spanning 100 time steps and meshes with up to 16K nodes. Although ROBIN was trained on BendingBeam with 3 hierarchies (~750 nodes), the shared blocks of the AMPNs enable direct application of ROBIN to BendingBeamLarge meshes with 4 levels. We fine-tuned ROBIN for 750K iterations with a batch size of 1 and a learning rate of $10^{-6}$. Similarly, we have trained an untrained ROBIN with the same settings, but with a learning rate decay ranging from $10^{-4}$ to $10^{-5}$.
>
> |  | **BendingBeamLarge** |  |
> | --- | --- | --- |
> | ROBIN variant | Rollout time [s] | Rollout RMSE [$10^{-3}$] |
> | pre-trained | 30.936 +/- 0.627 | 78.929 +/- 4.971 |
> | untrained | 30.945 +/- 0.580 | 215.500 +/- 12.135 |
>
> As expected, the pre-trained ROBIN model predicts solutions with a much lower RMSE than an untrained model. The pre-trained model can be quickly adapted to a large-scale dataset without architectural changes, which highlights ROBIN’s generalizability to different mesh sizes.
>
> > (Minor) The description of ROBI and corresponding figure could be improved; […] It took me a few passes to understand which step is conditioned on what exactly in diffusion & simulation time.
> >
>
> We thank the reviewer for pointing that out. We will clarify the description of ROBI, as well as Figure 2. In particular, we will clarify the conditioning process on the partially denoised, reconstructed state $\tilde{u}_{0|k}^{t-1}$ and explain how the reconstruction process of that state depends on previous time steps.
>
> We appreciate the positive and constructive feedback from the reviewer and will make the required updates in the final version. We are pleased to address any additional inquiries during the discussion phase.
>
> [1] Yadi Cao, et al. Efficient learning of mesh-based physical simulation with bi-stride multi-scale graph neural network. In International conference on machine learning. PMLR, 2023.

---

> > ### Comment · Reviewer_V9Hd · 2025-08-05
> >
> > Thank you for the additional experiments. I think it's definitely worth including those in the paper, or least the appendix-- I found especially the first experiment on truncated runs quite insightful (maybe with a figure, instead of a table).
> >
> > I think this is a good paper, and will retain my (positive) score.

---

> > > ### Author Response · Authors · 2025-08-07
> > >
> > > Dear reviewer,
> > >
> > > Thank you for your positive feedback and continued support of our work. We will follow your recommendation to include the additional experiments in the appendix and present the truncated runs as a figure.

---

### Official Review · Reviewer_WXuT · 2025-06-29

**Clarity:** 2
**Significance:** 3
**Originality:** 4
**Rating:** 5
**Confidence:** 4

**Summary:**

This work proposes an auto-regressive hierarchical graph neural network and diffusion model based framework for modelling the dynamics of systems in solid-mechanics from a Lagrangian frame. The paper extends the work of the paper PDE-Refiner Lippe et al. 2023 with several key extensions. They extend onto problems on unstructured meshes with Lagrangian dynamics using a hierarchical GNN. The mesh hierarchy utilises a novel AMG based approach from classical FE meshing, which creates visually more balanced and appealing coarse meshes compared to recent ML baselines. The authors also introduce a strategy Rolling Diffusion Batched Inference (ROBI), that allows accelerated roll-out at inference time by initialising the denoising process of future states conditioned by "early-denoising" of preceding states. Experiments are performed on 3 dynamical problems from solid mechanics, baselined against 2 SOTA models MGN and HCMT. A sensitivity analysis of the ROBI procedure wrt to $m$ the denoising stride and $K$ the denoising steps and a model ablation study of key architecture components are also performed.

**Questions:**

No

**Ethical Concerns:**

["NO or VERY MINOR ethics concerns only"]

**Final Justification:**

The provided clarifications and model inference timings have addressed my concerns.

**Limitations:**

Yes

**Quality:**

3

**Strengths And Weaknesses:**

This paper would be a straight accept for me if the quality of the write up was better but I had to spend quite some time to extract the details. In particular the clarity of the write up in sections 3.1 and 3.2 regarding ROBI and the HGNN could be improved, I list my concerns below to see they can be improved.

In section 3.1:
Regarding the v-prediction target, the explanation was not clear. My eventual understanding is when k=K (first denoising steps) this is where we sample pure noise which is input to the model but also the target is closest to the 1 step direct target. Conversely when k~0 (latter denoising steps) the target is the remaining noise on the sample. So when the SNR is the highest the target is noise and when the SNR is the lowest the target is the signal. I found this quite counter intuitive and perhaps it can be clarified/reworded. Also, "One step models" are not defined, this confused me for a long time until I read PDE-Refiner paper.

- There is some abiguity if there are also neural nets used to predict $\mu$ and $\Sigma$. But are these different to the main denoiser that predicts the denoising velocity and if so how are they trained?
- There are no definitions of t_B/k_B/t_b/k_b
- It was not immeadiate to derive the formula on line 148, it could be pointed out this comes from substituting $\epsilon$ from the noising step into 1.
- Clarifications on figure 3, what is meant by "pooling", as pooling is usually associated as a step within coarsening. In the caption does "1 (top)" refer to line c) ?

In the ROBI section, the writing would benefit from clarifying/reasserting that the network has 2 inputs, the t-1 condition and the current t sample at denoise level k, as this is non-standard in diffusion models, AR is a niche case. I now understand this is because the model takes in 2 arguments, previous step as condition and current noisey sample. It is the "early-denoising" of the preceding t-1 conditioning that gives a "trade-off" against accuracy when m<<K as it is utilised from a less developed state. Similarly, The sentence that starts "Using these properties, we begin", I thought for a long time the N(0,1) was a typo, given the prior sentence is motivating "sufficient to condition the next step".

In section 3.2:
The description of root-node AMG coarsening is also unclear, I would not be able to implement it from the description. Neither the rote-node or smoothing aspects from "algebraic root-node-based smooth aggregation" are explained. The description of upsampling/downsampling are not well explained "Those matrices are constructed by smoothing the sparse initial aggregation mapping using the adjecency matrix" I think I understand there is some kind of graph diffusion rewiring but providing more details would again strengthen the paper.

I have some small evaluation concerns. No numerical figures are reported just bar charts that are hard to read and interpret. For the rollout in Figure 5b, I understand we compare this is a sensitivity study for different $m$ and $K$ but it is a slight red flag that ML baseline and even classic data generation times are not provided. Especially since this is a diffusion model where inference time is a known challenge.

---

> ### Author Rebuttal · Authors · 2025-07-31
>
> > This paper would be a straight accept for me if the quality of the write up was better but I had to spend quite some time to extract the details. In particular the clarity of the write up in sections 3.1 and 3.2 […].
> >
>
> We thank the reviewer for the positive feedback and for identifying areas that can be improved in our write-up. We will revise sections 3.1 and 3.2 for the final version. Below, we address the reviewer’s questions and clarify several ambiguities.
>
> > Regarding the v-prediction target, the explanation was not clear. […] I found this quite counter intuitive and perhaps it can be clarified/reworded.
> >
>
> The v-prediction $\mathbf{v}_k^t = \sqrt{\bar{\alpha}_k} \mathbf{\epsilon}^t - \sqrt{1-\bar{\alpha}_k} \mathbf{u}^t_0$ indeed behaves in the opposite way to the SNR. At early denoising steps (low SNR, $\bar{\alpha}_k \approx0$), the target behaves like the clean signal $\mathbf{u}^t_0$. At later steps (high SNR, $\bar{\alpha}_k \approx1$), the target behaves like the residual noise **$\mathbf{\epsilon}^t$.** Although this may seem counterintuitive, it simplifies the learning process. When the sample is good (high SNR) correcting the small residual is easier than regenerating the entire signal. When the sample is random (low SNR), the input contains little useful structure, so directly predicting the signal is easier than estimating the error. This reparameterization also keeps target magnitudes more uniform across timesteps.
>
> > Also, "One step models" are not defined, this confused me for a long time until I read PDE-Refiner paper. […] There are no definitions of t_B/k_B/t_b/k_b.
> >
>
> We will introduce one-step models as autoregressive models that are trained to predict the solution for the next time step using the solution from the previous step. We will define $t_B$ and $k_B$ more clearly and use them consistently.
>
> > There is some abiguity if there are also neural nets used to predict $\mu$ and $\Sigma$. But are these different to the main denoiser that predicts the denoising velocity and if so how are they trained?
> >
>
> In Denosing Diffusion Probalistic Models [1] we train a model to predict either the clean sample $u_{0}^t$, the noise $\epsilon^t$ or, as in our case, the velocity $v^t$.  The Gaussian mean $\mu_{\theta}$ is than computed analytically from this output. The covariance matrix $\Sigma_{\theta}$ can also be learned by the same network, but we assume isotropic covariance $\Sigma = \sigma_k^2 \mathbf{I}$.
>
> > It was not immeadiate to derive the formula on line 148, it could be pointed out this comes from substituting from the noising step into 1.
> >
>
> Given a clean sample $u_0^t$ and noise $\epsilon^t$, the forward diffusion process produces at step $k$ a noisy sample $u_k^t=\sqrt{\bar\alpha_k}\,u_0^t+\sqrt{1-\bar\alpha_k}\,\epsilon^t$ [1]. We eliminate $\epsilon^t$ by substituting the v-prediction definition from Equation (1) $v_k^t=\sqrt{\bar\alpha_k}\,\epsilon^t-\sqrt{1-\bar\alpha_k}\,u_0^t$ and solve for the clean sample $u_0^t=\sqrt{\bar\alpha_k}\,u_k^t-\sqrt{1-\bar\alpha_k}\,v_k^t$ [1]. Given a model prediction $v_\theta(u_k^t,k,u_{0}^{t-1})$, we can reconstruct a clean sample $u_{0|k}^t=\sqrt{\bar\alpha_k}\,u_k^t-\sqrt{1-\bar\alpha_k}\,v_\theta(u_k^t,k,u_{0}^{t-1})$ using this equation.
>
> > Clarifications on figure 3, what is meant by "pooling", as pooling is usually associated as a step within coarsening. In the caption does "1 (top)" refer to line c) ?
> >
>
> In our AMG-based hierarchy, we create coarse meshes and build up- and downsampling graphs that connect nodes across levels. Figure 3 d) shows the edges between levels 0 and 1 at the top and between levels 1 and 2 at the bottom. All the nodes at the top correspond to level 0 (Figure a), while the subset of nodes marked in bright blue corresponds to level 1 (Figure c). Similarly, all the nodes in the bottom part of Figure 3d correspond to the nodes in Figure 3c. Since "pooling" is misleading, Figure 3 d) will be renamed "AMG-based down- and upsampling graphs.”
>
> > I understand we compare this is a sensitivity study for different and but it is a slight red flag that ML baseline and even classic data generation times are not provided.
> >
>
> The table below shows a comparison of Rollout RMSE between the baselines.
>
> |  | **BendingBeam** |  | **DeformingPlate** |  | **ImpactPlate** |  |
> | --- | --- | --- | --- | --- | --- | --- |
> |  | Rollout time [s] | Rollout RMSE [$10^{-3}$] | Rollout time [s] | Rollout RMSE [$10^{-3}$] | Rollout time [s] | Rollout RMSE [$10^{-3}$] |
> | MGN | 21.398 +/- 0.153 | 189.515 +/- 70.865 | 24.418 +/- 0.217 | 8.761 +/- 0.293 | 2.927 +/- 0.031 | 54.068 +/- 5.878 |
> | HCMT | 25.818 +/- 0.280 | 121.526 +/- 1.865 | 31.572 +/- 0.144 | 8.035 +/- 0.133 | 4.331 +/- 0.050 | 19.571 +/- 0.382 |
> | ROBIN | 15.026 +/- 0.040 | 29.002 +/- 1.997 | 61.601 +/- 0.291 | 4.978 +/- 0.332 | 4.943 +/- 0.010 | 12.330 +/- 0.958 |
>
> The numerical solver required 46.2 s (max. 186.1 s) on average on BendingBeam, 1157.2s on DeformingPlate [2] and 742.6 s on ImpactPlate [3].
>
> We also evaluated ROBIN on BendingBeamLarge (200 training, 20 validation and 20 test simulations), containing simulations of 100 time steps on beam meshes with 5–16K nodes. On average, the solver required 108.3 s (max. 4248.0 s) to complete. We compare the fine-tuning of a pre-trained model of BendingBeam to training from scratch. Both are trained for 750K iterations and a batch size of 1, while fine-tuning uses a learning rate of $10^{-6}$ and training from scratch an exponential decay from $10^{-4}$ to $10^{-5}$.
>
> |  | **BendingBeamLarge** |  |
> | --- | --- | --- |
> | ROBIN variant | Rollout time [s] | Rollout RMSE [$10^{-3}$] |
> | pre-trained | 30.936 +/- 0.627 | 78.929 +/- 4.971 |
> | untrained | 30.945 +/- 0.580 | 215.500 +/- 12.135 |
>
> We extended ROBI to include the option of truncating the denoising process after $K_I$ steps and using the partially denoised state as the final prediction. The table below shows for 5 seeds how different truncation levels $K_I$ compare to conventional inference and ROBIN (5 seeds). All evaluations are based on the same trained models.
>
> |  |  |  |  | **BendingBeam** |  | **DeformingPlate** |  | **ImpactPlate** |  |
> | --- | --- | --- | --- | --- | --- | --- | --- | --- | --- |
> | Inference variant |  $K$ | $K_I$ |  $m$ | Rollout time [s] | Rollout RMSE [$10^{-3}$] | Rollout time [s] | Rollout RMSE [$10^{-3}$] | Rollout time [s] | Rollout RMSE [$10^{-3}$] |
> | Conventional Inference | 20 | 20 | 20 | 162.426 +/- 0.891 | 28.896 +/- 1.906 | 190.100 +/- 1.333 | 4.960 +/- 0.355 | 22.607 +/- 0.358 | 14.431 +/- 1.933 |
> | ROBIN | 20 | 20 | 1 | 15.026 +/- 0.040 | 29.002 +/- 1.997 | 61.601 +/- 0.291 | 4.978 +/- 0.332 | 4.943 +/- 0.010 | 12.330 +/- 0.958 |
> | ROBIN - ten step | 20 | 10 | 1 | 11.030 +/- 0.099 | 30.349 +/- 2.409 | 33.705 +/- 0.098 | 5.181 +/- 0.347 | 2.896 +/- 0.009 | 12.379 +/- 0.923 |
> | ROBIN - five step | 20 | 5 | 1 | 9.879 +/- 0.188 | 35.812 +/- 3.299 | 20.243 +/- 0.038 | 5.729 +/- 0.299 | 1.907 +/- 0.011 | 13.011 +/- 0.977 |
> | ROBIN - two step | 20 | 2 | 1 | 9.475 +/- 0.105 | 48.936 +/- 7.248  | 12.588 +/- 0.127 | 7.263 +/- 0.179 | 1.424 +/- 0.005 | 15.788 +/- 1.714 |
> | ROBIN - one step | 20 | 1 | 1 | 9.488 +/- 0.117 | 64.232 +/- 11.862 | 7.113 +/- 0.042 | 10.787 +/- 0.521 | 1.324 +/- 0.006 | 21.462 +/- 2.115 |
>
>
>
> > In the ROBI section, the writing would benefit from clarifying/reasserting that the network has 2 inputs, the t-1 condition and the current t sample at denoise level k.
> >
>
> In the revision, we will state explicitly that at level $k$ during inference the model takes the current noisy sample $u_{k}^{t}$ and the partially denoised state of $\\tilde{u}\_{0|k}^{t-1}$ as input to predict the denoising velocity $v_\\theta(u\_k^t,k,\\tilde{u}\_{0|k}^{t-1})$. We will clarify that the refinement level of the reconstructed sample $\\tilde{u}_\{0|k+1-m}^{t-1}$ increases with $m$.
>
> > The description of root-node AMG coarsening is also unclear […] but providing more details would again strengthen the paper.
> >
>
> We use PyAMG’s ````rootnode_solver```` [4] to create the AMG hierarchy. The solver takes a sparse square matrix as input and returns a hierarchical cycle. We pass the fine-mesh adjacency matrix as the system matrix, and the solver yields coarsened adjacency matrices, corresponding coarse nodes, and up- and downsampling matrices. The nonzeros in these matrices define the edges of our up- and downsampling graphs.  Thus, the implementation requires only the fine-mesh adjacency, the ````rootnode_solver````  with its default settings, and graph construction based on the returned matrices (i.e., non-zero values).
>
> The ````rootnode_solver```` itself aggregates nodes based on connectivity within the provided matrix and selects one root per aggregate. These roots become the next-level coarse nodes, and each fine node connects to its aggregate's root in the up- and downsampling matrices. These operators are then smoothed using the adjacency, effectively enlarging each coarse node’s receptive field beyond its aggregate. The implementation will be included in the code publication after the paper is accepted.
>
> We thank the reviewer for their comments and for pointing out the ambiguities. We will incorporate the revsions into the camera-ready version, and we encourage the reviewer to contact us during the discussion.
>
> [1] Jonathan Ho, et al. Denoising diffusion probabilistic models. NeurIPS, 2020.
>
> [2] Tobias Pfaff, et al. Learning Mesh-Based Simulation with Graph Networks. ICLR, 2020.
>
> [3] Youn-Yeol Yu, et al. Learning Flexible Body Collision Dynamics with Hierarchical Contact Mesh Transformer. ICLR, 2023.
>
> [4] Nathan Bell, et al. PyAMG: Algebraic multigrid solvers in python. Journal of Open Source Software, 2022

---

> > ### Comment · Reviewer_WXuT · 2025-08-05
> >
> > I thank the authors for the clarifications, which I am confident can be polished in the final transcript.
> > Thank you for providing the timings, this is particularly encouraging that the model inference time (in part thanks to ROBIN) is comparable to non-diffusion baselines.
> > I update my score accordingly.

---

> > > ### Author Response · Authors · 2025-08-07
> > >
> > > Dear Reviewer,
> > >
> > > Thank you for your encouraging feedback and updated assessment. We're glad our clarifications were helpful and appreciate your valuable comments and feedback, which we will incorporate into the final version.

---

### Official Review · Reviewer_GHwE · 2025-07-01

**Clarity:** 2
**Significance:** 2
**Originality:** 3
**Rating:** 4
**Confidence:** 4

**Summary:**

Graph based simulators are widely used for the mesh based data modalities, however these models struggle with long range dependencies and correlations. Furthermore autoregressive rollouts leads to error accumulation which introduces another source of error. This paper introduces ROBIN a diffusion based learned simulator to address these issues. A rolling diffusion denoising scheme is introduced to accelerate the overall generation process by parallelizing the diffusion across time. Furthermore a hierarchical graph neural network built on multigrid coarsening is introduced to allow for multiscale message passing.

**Questions:**

1) Can authors provide a clearer picture of the novelty and comparison with existing autoregressive diffusion methods? It is not clear how the rollout denoising differs from others!

2) If it is possible to put the dependence of the denoising between steps into equation it would become much more clear to the reader. Can authors formulate such equation (Even if it is a small inline equation)?

3) Can authors clarify why a parallel generation is not possible in this concept? This way the complexity simply becomes O(T)!

4) Why did the authors focus on a very limited area? Is it possible to expand this to other PDEs? This will help understanding the generalizability.

**Ethical Concerns:**

["NO or VERY MINOR ethics concerns only"]

**Final Justification:**

Authors provided further comparison with other frameworks demonstrating improved rollout. Furthermore the denoising scheme is novel in this context. I believe the experiments are done on a very specific subset of the PDEs, and generalizability is still of concern however given the novelties I believe the reasons to accept the paper are marginally more than the reasons to reject.

**Limitations:**

Yes

**Paper Formatting Concerns:**

No concerns

**Quality:**

3

**Strengths And Weaknesses:**

**Strength**
1) The concept of a rollout diffusion inference and autoregressive diffusion is rapidly growing popularity for world models and other scenarios where actions and inputs are continuously introduced to the model. The rollout diffusion idea of the paper is interesting to see in the context of solid mechanics.

2) The multi-resolution message passing opens doors to mixing simulations at different scales as well. This is something that can be explored by other researchers to have a generations informed from variety of scales.


**Weakness**
1) A major concern is the time parallel denoising scheme for inference. Authors mention that for generation of T frames (timesteps) with K denoising steps the time complexity will be O(KT). However the first diffusion models introduced for spatiotemporal data were not trained or used at inference with rollouts. These models (and most current video generative models) denoise all frames (timesteps) in parallel with an inflated UNet so the actual complexity would be O(K). It seems the paper uses the argument of efficiency throughout to develop the the rollout style denoising scheme to achieve O(K-m + mT).

2) Paper provides the arguments to how they get to O(K-m + mT) in "Rolling Diffusion-Batched Inference (ROBI)" section however their claim in introduction is that they reduce the inference time from O(KT) to O(T)

3) Novelty of the ROBI is unclear. How does ROBI differ from the current autoregressive diffusion frameworks.

4) Provided experiments are limited. Given the claim of a new denoising scheme is introduced authors potentially should provide comparisons with other autoregressive diffusion frameworks (mostly in image or video generation). Otherwise there is a wide range of PDE problems that are more commonly used for benchmarking that can be used to compare methods with baselines.

5) Baselines are old and and limited. There has been major researches focusing on PDE generations in recent years however the paper only compares to two baselines one of which is older than 5 years. MGN is commonly is not used as a standalone baseline and it helps complete an overall picture when also having other baselines.

---

> ### Author Rebuttal · Authors · 2025-07-31
>
> We thank the reviewer for their careful reading and constructive criticism. Below, we clarify our contributions and explain how they differ from existing work.
>
> > Can authors provide a clearer picture of the novelty and comparison with existing autoregressive diffusion methods?
> >
>
> ROBIN is designed for deterministic physical simulations rather than perceptual video generation. It combines a diffusion model with hierarchical graph neural networks on unstructured meshes. This allows it to capture different solution frequencies and propagate information across large spatial distances and mesh resolutions. CNN-based video models cannot do this. In addition, ROBIN outputs residual predictions rather than absolute states, thereby improving accuracy and stability.
>
> ROBIN advances the simulation from time $t−1$ to time $t,$ conditioned only on the previous clean state. This preserves causality and time shift equivariance while reducing memory usage and training time.  Sequence-based diffusion models that denoise entire histories often require learning warm-up and cool-down phases [1]. Conditioning on multiple past steps has been found to reduce the accuracy of learned simulators [2,3].
>
> Our model is trained using a v-prediction target, residual predictions and an exponential diffusion scheduler. It requires only 5 diffusion steps to surpass the baseline accuracy. During the rebuttal, we evaluated inference truncation. We stop after a chosen number of denoising steps and use the reconstructed sample as the prediction, which decreases inference time. The table below reports (over 5 seeds) on variants that stop denoising after the second denoising step (ROBIN - two step), along with the full ROBIN setting and the baselines. We added Bi-stride multi-scale GNNs (BSMS-GNNs) [4] as a third baseline on BendingBeam and will include results for DeformingPlate and ImpactPlate in the camera-ready version.
>
> |  | **BendingBeam** |  | **DeformingPlate** |  | **ImpactPlate** |  |
> | --- | --- | --- | --- | --- | --- | --- |
> |  | Rollout time [s] | Rollout RMSE [$10^{-3}$] | Rollout time [s] | Rollout RMSE [$10^{-3}$] | Rollout time [s] | Rollout RMSE [$10^{-3}$] |
> | ROBIN - two step | 9.475 +/- 0.105 | 48.936 +/- 7.248  | 12.588 +/- 0.127 | 7.263 +/- 0.179 | 1.424 +/- 0.005 | 15.788 +/- 1.714 |
> | ROBIN | 15.026 +/- 0.040 | 29.002 +/- 1.997 | 61.601 +/- 0.291 | 4.978 +/- 0.332 | 4.943 +/- 0.010 | 12.330 +/- 0.958 |
> | MGN | 21.398 +/- 0.153 | 189.515 +/- 70.865 | 24.418 +/- 0.217 | 8.761 +/- 0.293 | 2.927 +/- 0.031 | 54.068 +/- 5.878 |
> | HCMT | 25.818 +/- 0.280 | 121.526 +/- 1.865 | 31.572 +/- 0.144 | 8.035 +/- 0.133 | 4.331 +/- 0.050 | 19.571 +/- 0.382 |
> | BSMS-GNNs | 4.896 +/- 0.124 | 141.979 +/- 7.960 | - | - | - | - |
>
> ROBIN outperforms state-of-the-art baselines with only two diffusion steps, whereas video diffusion models typically require many more. While extending ROBI to video models is an interesting future direction, our models are explicitly designed for simulation. We do not claim to have made any contributions toward overcoming the limitations of current image or video generation frameworks.
>
> > There has been major researches focusing on PDE generations in recent years however the paper only compares to two baselines one of which is older than 5 years.
> >
>
> As noted above, we added BSMS‑GNNs [4] as another strong baseline. HCMT currently achieves the highest accuracy on benchmarks such as DeformingPlate, and MeshGraphNet (MGN), despite being five years old, remains competitive. This demonstrates the importance of ROBIN in surpassing the accuracy limits of solid-mechanics learned simulators.
>
> > Authors mention that for generation of T frames (timesteps) with K denoising steps the time complexity will be O(KT). […] Can authors clarify why a parallel generation is not possible in this concept?
> >
>
> ROBIN generates trajectories autoregressively, predicting each step based on the previous one. Forecasting large time steps or an entire rollout at once reduces the accuracy of physical simulators [2,5]. The most accurate solid‑mechanics simulators, such as HCMT, are autoregressive.
>
> Since physical simulations are Markovian, conditioning only on the previous step is sufficient. Jointly denoising all time steps ignores the evolving state and violates causality. Even using more than one past step can reduce accuracy [2,3]. Additionally, parallel generation is less desirable because sequence models must retain many time steps and large meshes in memory, treating entire trajectories as single training samples. Our one step training uses single transitions, which improves memory and data efficiency while enabling large-mesh experiments.
>
> We demonstrate that scalability on BendingBeamLarge, which contains 200 training, 20 validation and 20 test simulations of 100 time step and beam meshes with 5K-16K nodes. We fine-tuned a pre-trained model and compare it with a model trained from scratch (over 5 seeds). Both used 750K iterations and a batch size of 1. The pre-trained model is trained with a learning rate of $10^{-6}$ and the untrained with an exponential decay from $10^{-4}$ to $10^{-5}$.
>
> |  | **BendingBeamLarge** |  |
> | --- | --- | --- |
> | ROBIN variant | Rollout time [s] | Rollout RMSE [$10^{-3}$] |
> | pre-trained | 30.936 +/- 0.627 | 78.929 +/- 4.971 |
> | untrained | 30.945 +/- 0.580 | 215.500 +/- 12.135 |
>
> Pretraining substantially improves accuracy and enables adaptation to meshes with up to 16K nodes.
>
> > Paper provides the arguments to how they get to O(K-m + mT) in "Rolling Diffusion-Batched Inference (ROBI)" section however their claim in introduction is that they reduce the inference time from O(KT) to O(T)
> >
>
> ROBI reduces inference costs from $O(KT)$ to $O(K-m + mT)$. Our experiments set $m=1$, yielding $O(K + T)$. Since time‑dependent simulations typically have many more time steps $T$ than the $K=5–20$ diffusion steps , this simplifies to  $O(T)$. We will clarify this assumption in the camera‑ready version.
>
> > If it is possible to put the dependence of the denoising between steps into equation it would become much more clear to the reader.
> >
>
> Given a prediction horizon $h=K$, a batch of previous clean states $\mathbf{u}^{t_j-1}\_{0}$ and noise samples  $\Delta\mathbf{u}^{t_j}\_{k_j}$ for the time steps $t_j = t+j$ and $j\in\{0,...,h-1\}$ with increasing diffusion steps $k_j=1+j$, ROBIN outputs the diffusion velocities $\mathbf{v}\_\theta(\Delta\mathbf{u}^{t_j}\_{k_j}, k_j, \mathbf{u}^{t_j-1}\_0)$.
> We approximate the clean samples $\Delta\tilde{\mathbf{u}}^{t_j}\_{0} \approx \Delta\tilde{\mathbf{u}}^{t_j}\_{0|k_j} = \sqrt{\bar{\alpha}\_{k_j}} \Delta\mathbf{u}^{t_j}\_{{k_j}} - \sqrt{1-\bar{\alpha}\_{k_j}} \mathbf{v}\_\theta(\Delta\mathbf{u}^{t_j}\_{{k_j}}, {k_j}, \mathbf{u}^{t_j-1}\_0)$  and the current noisy samples $\\Delta\\mathbf{u}^{t\_j}\_{k\_j-1}=\\frac{\\sqrt{\\bar{\\alpha}\_{k\_j-1}}{\\beta}\_{k\_j}}{1-\\bar{\\alpha}\_{k\_j}}\\Delta\\tilde{\\mathbf{u}}^{t\_j}\_{0\\mid k\_j} +
> \\frac{\\sqrt{\\alpha\_{k\_j}}(1-\\bar{\\alpha}\_{k\_j-1})}{1-\\bar{\\alpha}\_{k\_j}} \\Delta\\mathbf{u}^{t\_j}\_{k\_j}
>  $ $ + \\sigma\_{k\_j} \\boldsymbol{\\varepsilon}^{t\_j} $ with $ \\boldsymbol{\\varepsilon}^{t\_j}\\sim\\mathcal{N}(0,I) $. Next, we can reconstruct the states of each time step using a cumulative sum over those approximated clean samples $\\tilde{\\mathbf{u}}^{{t_j}}\_{0} = \\mathbf{u}^{t-1}\_{0} + \\sum\_{i=0}^j \\Delta \\tilde{\\mathbf{u}}^{t+i}\_{0}$ with the last fully-denoised sample $\\mathbf{u}^{t-1}\_{0}$. Finally, we can move one physical time step further, such that  $t_j = t+1+j$. Therefore, we remove the denoised time step $t$ from the batch, initialize the sample of the new time step ${t+h}$ as Gaussian noise $\\Delta\\mathbf{u}^{t+h}\_{K} \\sim \\mathcal{N}(\\mathbf{0}, \\mathbf{I})$ and predict the next batch $\\mathbf{v}_\\theta(\\Delta\\mathbf{u}^{{t\_j}}\_{k\_j}, {k\_j}, \\tilde{\\mathbf{u}}^{{t\_j}-1}\_{0} = \\mathbf{u}^{t}\_{0} + \\sum\_{i=1}^j \Delta \\tilde{\\mathbf{u}}^{t+i}\_{0})$. Note that the proposed state reconstruction is done before the model call and allows the model to treat all time steps as a batch, where each prediction is only conditioned on the previous reconstructed state, e.g., $\\mathbf{v}\_\\theta(\\Delta\\mathbf{u}^{t+2}\_2, 2, \\tilde{\\mathbf{u}}^{t+1}\_{0})$ for time step $j=1$.
>
> > Why did the authors focus on a very limited area? Is it possible to expand this to other PDEs?
> >
>
> The present work focuses on solid mechanics, a significant engineering discipline encompassing manufacturing, structural mechanics, and lightweight design, where accurate learned simulation remains under-explored. Solid mechanics exhibit global dependencies and high solution frequencies, which present a challenge for learned simulators. However, ROBIN can handle both by effectively combining diffusion with hierarchical GNNs. Although this study is limited to solid mechanics, the method could be applied to other PDEs, including fluid dynamics. Diffusion and hierarchical CNN‑based simulators improve image‑based fluid simulations [1,2]. We plan to apply ROBIN to mesh-based fluid simulations in future work.
>
> We thank the reviewer for the suggestions and hope that we have addressed all concerns. We will include the new results in the revised paper and welcome further questions during the discussion phase.
>
> [1] David Ruhe, et al. Rolling Diffusion Models. ICML, 2024.
>
> [2] Phillip Lippe, et al. PDE-Refiner: Achieving Accurate Long Rollouts with Neural PDE Solvers. NeurIPS, 2023.
>
> [3] Tobias Pfaff, et al. Learning Mesh-Based Simulation with Graph Networks. ICLR, 2020.
>
> [4] Yadi Cao, et al. Efficient learning of mesh-based physical simulation with bi-stride multi-scale graph neural network. PMLR, 2023.
>
> [5] Zongyi Li, et al. Learning Chaotic Dynamics in Dissipative Systems. NeurIPS, 2022.
>
> [6] Kiwhan Song, et al. History-Guided Video Diffusion. *arXiv*, 2025.

---

> ### Author Response · Authors · 2025-08-08
>
> Dear reviewer,
>
> Thank you for your thoughtful and detailed feedback. We hope that our response has clarified the concerns you raised and addressed the specific points you noted. Given the short discussion period, we would be grateful to know whether our responses have resolved your concerns or if there are any other areas in which we could improve the revision.
>
> We value your consideration and the opportunity to strengthen the paper.

---

### Official Review · Reviewer_bZkb · 2025-07-07

**Clarity:** 3
**Significance:** 3
**Originality:** 3
**Rating:** 5
**Confidence:** 2

**Summary:**

The authors proposed a novel method, Rolling Diffusion Batched Inference (ROBI), that can accelerate inference in DDPM-based simulation, and then they combined it with a Hierarchical Graph Neural Network that combines multiscale message passing with ROBI to provide fast, accurate diffusion-based simulations.

**Questions:**

1. How is the computation cost compared to HGNN? Does it take longer or shorter to train？
2. How does ROBI learn the parameter compared to conventional diffusion?

**Ethical Concerns:**

["NO or VERY MINOR ethics concerns only"]

**Limitations:**

yes

**Quality:**

3

**Strengths And Weaknesses:**

The strengths of this paper are:
1. The paper has a clear model structure and mathematical formula which reinforce the theoretical soundness and reliability of the proposed model.
2. The model's novelty is good, although many people are using the Diffusion model, the authors improve it here.
3. The paper features a detailed model structure that is easy to follow and understand, and the experiment is well-designed, clear, and straightforward.

---

> ### Author Rebuttal · Authors · 2025-07-31
>
> We appreciate the thorough and favorable review, especially the recognition of our meticulous mathematical formulation and well-designed experimental setup. Below, we will briefly address the questions and concerns raised.
>
> > How is the computation cost compared to HGNN? Does it take longer or shorter to train？
> >
>
> We agree that it is important to report the computation cost against the baselines. Previously, we only presented ROBIN runtimes for various diffusion inference settings in Figure 5. We report the rollout time   and RMSE of ROBIN and the baselines below. We also added an option to truncate the denoising process to speed up ROBIN’s inference further. In this setting, we stop after a chosen number of denoising steps, treating the reconstructed sample as the prediction and skipping the remaining steps. We report on variants that stop after the first (ROBIN - one step), second (ROBIN - two step), and the fifth (ROBIN - five step) steps.
>
> |  | **BendingBeam** |  | **DeformingPlate** |  | **ImpactPlate** |  |
> | --- | --- | --- | --- | --- | --- | --- |
> |  | Rollout time [s] | Rollout RMSE [$10^{-3}$] | Rollout time [s] | Rollout RMSE [$10^{-3}$] | Rollout time [s] | Rollout RMSE [$10^{-3}$] |
> | ROBIN - one step | 9.488 +/- 0.117 | 64.232 +/- 11.862 | 7.113 +/- 0.042 | 10.787 +/- 0.521 | 1.324 +/- 0.006 | 21.462 +/- 2.115 |
> | ROBIN - two step | 9.475 +/- 0.105 | 48.936 +/- 7.248  | 12.588 +/- 0.127 | 7.263 +/- 0.179 | 1.424 +/- 0.005 | 15.788 +/- 1.714 |
> | ROBIN - five step | 9.879 +/- 0.188 | 35.812 +/- 3.299 | 20.243 +/- 0.038 | 5.729 +/- 0.299 | 1.907 +/- 0.011 | 13.011 +/- 0.977 |
> | ROBIN | 15.026 +/- 0.040 | 29.002 +/- 1.997 | 61.601 +/- 0.291 | 4.978 +/- 0.332 | 4.943 +/- 0.010 | 12.330 +/- 0.958 |
> | MGN | 21.398 +/- 0.153 | 189.515 +/- 70.865 | 24.418 +/- 0.217 | 8.761 +/- 0.293 | 2.927 +/- 0.031 | 54.068 +/- 5.878 |
> | HCMT | 25.818 +/- 0.280 | 121.526 +/- 1.865 | 31.572 +/- 0.144 | 8.035 +/- 0.133 | 4.331 +/- 0.050 | 19.571 +/- 0.382 |
>
> A key advantage of ROBIN is that our AMPNs are trained as one step autoregressive models that are conditioned only on the previous time step. We observed no additional training time compared to conventional autoregressive models. All methods used a fixed two-day training budget.
>
> Furthermore, we conducted an upscaling experiment that demonstrated ROBIN’s architecture is independent of mesh size. This means that, for larger mesh sizes, the number of layers in our AMG-based hierarchy increases, and the AMPNs can handle them because of the shared model blocks. We fine-tuned the pre-trained ROBIN models using the new BendingBeamLarge dataset. While the BendingBeam dataset contains meshes with an average of 750 nodes, the large dataset contains an average of 9K nodes, with a maximum of 16K nodes. We trained both the pre-trained and untrained models for 750K training iterations with a batch size of $1$. We fine-tuned the pre-trained models with a learning rate of $10^{-6}$, and we used a learning rate decay ranging from $10^{-4}$ to $10^{-5}$ for the untrained models.
>
> |  | **BendingBeamLarge** |  |
> | --- | --- | --- |
> | ROBIN variant | Rollout time [s] | Rollout RMSE [$10^{-3}$] |
> | pre-trained | 30.936 +/- 0.627 | 78.929 +/- 4.971 |
> | untrained | 30.945 +/- 0.580 | 215.500 +/- 12.135 |
>
> Pretrained models converge quickly on meshes with about twelve times more nodes, which highlights the ability of ROBIN to upscale and its training efficiency in fine tuning.
>
> > How does ROBI learn the parameter compared to conventional diffusion?
> >
>
> We thank the reviewer for the insightful question. ROBI is an inference rollout scheme, and its training is identical to that of conventional diffusion-based inference. It does not have any additional parameters that need to be learned.
>
> ROBI intuitively uses the insight that each diffusion step progressively predicts higher-frequency features to already start denoising future time steps before the current step has been fully denoised. Although this process does not reduce the number of required diffusion steps, it enables the parallel denoising of multiple steps. Figure 2 provides a schematic overview, and Figure 5 evaluates this effect. Empirically, we find that ROBI retains the accuracy of conventional diffusion inference while substantially speeding up the process on accelerated hardware. We will add a brief paragraph clarifying this relationship to the paper.
>
> We want to thank the reviewer again for the helpful feedback. We hope that our clarifications and new results address the concerns raised. We encourage the reviewer to reach out to us during the discussion if there are further questions.

---

> > ### Author Response · Authors · 2025-08-08
> >
> > Dear reviewer,
> >
> > Thank you for your positive and constructive feedback. We are grateful for your engagement with our work and for the encouraging comments you provided. If you have any further suggestions that could help to refine the paper, we would be glad to take them into account when revising it.
> >
> > We are grateful for your consideration and support.

---

### Note · Authors · 2025-08-15

We appreciate the reviewers' positive and constructive feedback. During the rebuttal, we conducted additional experiments to address the key concern and further strengthen the paper:

- **Baselines and architecture.** We have added Bi-stride multi-scale GNNs (BSMS-GNNs) on BendingBeam as an additional robust hierarchical GNN baseline. However, ROBIN is still significantly more accurate than all baselines. We also replaced the AMPNs in ROBIN with the HCMT architecture. While this ROBIN-HCMT variant improves on HCMT alone, ROBIN with AMPNs still performs best. This demonstrates that the improvements in ROBIN stem from the combination of diffusion and AMPNs rather than diffusion alone.
- **Runtime.** Direct runtime comparisons have been added against the baselines MGN, HCMT, and BSMS-GNNs, and against conventional numerical solvers. Thanks to Rolling Diffusion-Batched Inference (ROBI), ROBIN's runtime is comparable to the baselines despite its iterative diffusion process. ROBIN is also significantly faster than conventional numerical solvers. We introduced truncated inference to ROBIN, which stops denoising after a chosen number of steps to further increase inference speed. The first two diffusion steps are sufficient to surpass the accuracy of all baselines across all datasets. Varying the number of steps provides another controllable trade-off between inference speed and accuracy.
- **Frequency behavior.** We evaluated the RMSE of displacement and displacement gradient for different numbers of truncated diffusion steps and denoising strides. The results demonstrate that the initial diffusion steps recover the low-frequency global structure. Later steps refine high-frequency details. ROBIN achieves the same level of accuracy as conventional diffusion-based inference for both displacement and gradient, while being much faster.
- **Generalization to large meshes.** We added the BendingBeamLarge dataset, which contains meshes with up to 16K nodes. Our results demonstrate that fine-tuning a pretrained ROBIN model converges much faster than training from scratch. This demonstrates the ability to scale to larger meshes and larger AMG hierarchies without requiring architectural changes.

We think that these results reinforce our contributions and we will incorporate them into the revised manuscript. We thank the reviewers for their engagement and for helping to strengthen the paper.

---

### Decision · Program_Chairs · 2025-09-17

**Decision:**

Accept (spotlight)

**Comment:**

This paper proposes an improved diffusion model based on HGNN to simulate solid mechanics, which can improve both the quality and the speed.

Strengths:
1. The method is novel and has many interesting ideas, even it is based on some existing work.
2. The results are quite convincing and cover a good range of solid mechanics.

Weakness:
1. The initial draft lacks some detailed discussions.

Rebuttal:
Both the authors and reviewers actively participated in discussion. The authors provided extensive new results. The reviewers were very positive of the rebuttal, and raised the points.

Justification of recommendation:
A solid paper with novel ideas. Very strong rebuttal. Please include the new results and address the final comments in the final version.